manuscript submitted to *Atmospheric Measurement Techniques*

# Assimilation of lidar planetary boundary layer height observations.

**Andrew Tangborn[1], Belay Demoz[1,2], Brian J. Carroll[2], Joseph Santanello[3]and**

**Jeffrey L. Anderson[4]**

[1]Joint Center for Earth Systems Technology, University of Maryland Baltimore County, Baltimore, MD,

USA

[2]Dept. of Physics, University of Maryland Baltimore County, Baltimore, MD, USA

[3]Hydrological Sciences Laboratory, NASA Goddard Space Flight Center, Greenbelt, MD, USA

[4]National Center for Atmospheric Research, Boulder, CO, USA

Corresponding author: Andrew Tangborn, `tangborn@umbc.edu`

**Abstract**

Lidar backscatter and wind retrievals of the planetary boundary layer height (PBLH) are assimilated into 22 hourly forecasts from the NASA Unified - Weather and Research Forecast (NU-WRF) model during the Plains Elevated Convection Convection at Night (PECAN) campaign on July 11, 2015 in Greensburg, Kansas, using error statistics collected from the model profiles to compute the necessary covariance matrices. Two separate forecast runs using different PBL physics schemes were employed, and comparisons with 6 independent radiosonde profiles were made for each run. Both of the forecast runs accurately predicted the PBLH and the state variable profiles within the planetary boundary layer during the early morning, and the assimilation had a small impact during this time. In the late afternoon, the forecast runs showed decreased accuracy as the convective boundary layer developed. However, assimilation of the Doppler lidar PBLH observations were found to improve the temperature and V velocity profiles relative to independent radiosonde profiles. Water vapor was overcorrected, leading to increased differences with independent data. Errors in the U velocity were made slightly larger. The computed forecast error covariances between the PBLH and state variables were found to rise in the late afternoon, leading to the larger improvements in the afternoon. This work represents the first effort to assimilate PBLH into forecast states using ensemble methods.

## 1 Introduction

The planetary boundary layer (PBL) plays an important role in weather, climate and pollution through its role in land-atmosphere interactions and mediation of Earth's water and energy cycles (Santanello et al. 2018). This layer is where the Earth's surface interacts with the atmosphere, exchanging momentum, heat, moisture and pollutants. The PBL height (PBLH) is central to these interactions and is controlled by the energy flux from the surface. Under certain conditions during daytime it defines the convective boundary layer (CBL) and during nighttime it is the stable (non-convective) boundary layer (SBL). Trace gases and aerosols emitted from the surface are rapidly transported within the CBL by turbulent atmospheric motion, and transfer of energy and mass into the free troposphere occurs across an interfacial layer at the top of the PBL. The PBL affects convection in the troposphere, which is generally initiated within the boundary layer and then penetrates its top (Hong and Pan, 1998; Browning, et al. 2007). Thus,

accurate knowledge of the PBLH is essential for both weather, pollution and climate forecasting.

The PBLH is defined by thermodynamic properties such as a temperature inversion or hydrolapse which can be measured by radiosonde. Alternatively, the drop off in aerosol concentration that occurs across the top of the PBL is used, since aerosols are well mixed throughout the PBL when the CBL is present (Hicks, et al., 2019). Atmospheric models rely on parameterization schemes to define the structure of the PBL and compute PBLH. These are generally either local mixing schemes that use local turbulent kinetic energy (TKE, Janjic, 1994) or non-local flux schemes (Hong and Pan, 1996). Generally, these PBL parameterizations have systematically higher PBLH relative to observed values (Hegarty et al., 2018), and also have difficulties modeling the growth of the convective layer during the morning. The variety of definitions of PBLH make it difficult to effectively evaluate existing models or develop new ones.

Observations of PBLH are traditionally made by radiosonde measurements, which have high vertical resolution but are expensive to launch frequently and are thus limited to special experiments and/or ill-timed launches (*e.g.* 00/12 UTC National Weather Service launches) with respect to convective and stable PBL development. Likewise, spaceborne measurements of the lower troposphere from passive and active instruments are severely limited in vertical, spatial, and/or temporal resolution (Wulfmeyer et al. 2015). Ground based measurement of PBLH has been proposed for an extensive network of ceilometers by adding to the functionality of instruments that were designed for measuring cloud heights (Hicks et al., 2016). The ceilometer measures the time required for a laser pulse to return to a receiver, from which the height of the scattering is determined. The intensity of the backscatter is correlated with the density of aerosols at a given height and the PBLH is inferred from the location of the maximum negative gradient of the backscatter intensity. Several algorithms employ wavelet transforms to identify the location of the negative gradient (e.g. Brooks, 2003; Knepp, *et al.*, 2017). This existing network of ceilometers could be used to create a relatively dense network of frequent PBLH observations, as was recommended by the 2009 study from the National Research Council (NRC, 2009) and the Thermodynamic Profiling Technologies Workshop (NCAR, 2012).

Since the ceilometer PBLH observations were not yet available for the time period we are studying, we employ Doppler lidar observations made at the Plains Elevated Con-

vection at Night (PECAN) site in Greensburg, Kansas, to demonstrate the methodology. PECAN was an intensive campaign to study organized Mesoscale convection systems (MCSs) during the period June 1-July 15, 2015. It employed three aircraft and a large array of ground based lidar, radar and ground weather stations. The data we are using is from a Leosphere WINDCUBE-200S Doppler lidar owned and operated by the University of Maryland, Baltimore County (Delgado et al., 2016). This lidar operates at an infrared wavelength, and hence receives its strongest backscattered signal within the aerosol-laden PBL and is often below the measurement noise floor above the PBL. The Doppler shift of the backscattered signal is used to calculate wind speed as a function of range, which can then be used to produce a multitude of wind and turbulence variables useful for PBL characterization (e.g. vertical velocity variance and signal-to-noise ratio variance). While Doppler lidars and ceilometers are similar in aerosol detection, a Doppler lidar's additional wind measurement capability makes it more broadly applicable and at times more accurate than a ceilometer for PBLH retrievals. The PBLH algorithm applied for this study combines several such aerosol and wind variables and each PBLH retrieval involves measurement of turbulence intensity, horizontal wind profiles and backscatter intensity. The heights of steep gradients in these quantities are determined using empirical thresholds and wavelet transform techniques, and the three estimates are combined using fuzzy logic. This is described at length in Bonin et al. (2018). Additional lidar parameters and the application of the algorithm to PECAN data were presented in Carroll et al. (2019). The PBLH retrievals were made from a repeating 25-minute lidar scan cycle. This Doppler lidar and PBLH algorithm combination are generally well-suited for accurate and precise measurement of the PBLH during the daytime boundary layer, nocturnal boundary layer, and morning transition period (Bonin et al. 2018, Carroll et al. 2019). The evening transition is the most challenging for this setup due to due to difficulties in defining a clear mixing layer during the decay of a turbulent daytime PBL (Lothon et al. 2014).

The question remaining is how to assimilate these observations into a numerical weather prediction (NWP) model. A number of studies have explored assimilating boundary layer wind profile measurements from lidar (Hu et al. 2019, Coniglio et al. 2019, Degelia et al. 2019) and have shown that this increases the accuracy of forecasts due to improvements within the PBL. And further studies (Degelia et al. 2020; Chipilski et al. 2020) found that convective initiation (CI) was enhanced through the assimilation of thermo-

dynamic profiles within the PBL, though the former found that CI was degraded by the assimilation of kinematic (velocity) profiles. This work highlights the important role that the PBL plays in forecasting convective events, so that any observations that can improve estimation of the model state should be an important source of new information. We are interested assimilating the PBLH observations directly because the ceilometer network described above will focus on these retrievals, and satellite missions which measure PBLH are also planned. PBLH is a diagnostic variable in NWP parameterized physics models. This means any correction to PBLH will be lost during the model forecast unless the PBLH height observation is used to correct state variables such as temperature and moisture. This could be done either by adopting a variational data assimilation scheme, or through the use of an ensemble Kalman filter which would determine the error covariances between PBLH and state variables in the model. We choose the latter so as to avoid the task of linearizing the model physics. The structure of the covariance, and how the state variables are changed by assimilating PBLH, will depend on which PBL scheme is used. We will show how such a system could work by conducting a posteriori lidar PBLH observation impact experiments using forecast fields from a NASA Unified - Weather and Research Forecast (NU-WRF, Lidard-Peters, 2015) model runs for one day during the Plains Elevated Convection at Night (PECAN) campaign on July 11, 2015. The assimilation is done on 22 hourly WRF forecast fields throughout the day without cycling the analysis fields back into the model, using two different PBL parameterizations. In this paper, we demonstrate a new and promising method that uses the lidar-based aerosol backscatter and wind derived PBLH to correct model forecasted state variables. The purpose here is to show how ensemble computed error covariance can transfer observational information from PBLH to the state variable profiles.

## 2 Methodology

The assimilation methodology is based on the ensemble Kalman filter (EnKF)(Evenensen, 1994; Burgers, et al. 1998; Evensen, 2009), where the analysis state is the estimate with a minimized error norm, relative to the given error statistics. It differs from the EnKF in that the analysis is not used as an initial state for the next model forecast. Rather, two existing one day NU-WRF forecasts, with different PBL physics schemes, are used when lidar measurements are available at a single location. These forecasts were produced as a part of the PECAN campaign in 2015, and we reuse them here to demonstrate the

assimilation algorithm that we have developed. These were not ensemble forecasts so we cannot build a standard ensemble Kalman filter from them. Instead we use Ensemble Optimal Interpolation (EnOI), in which profiles from neighboring model gridpoints are used to obtain an estimate of error statistics (Oke, *et al.*, Evensen, 2003; 2010; Keppenne, *et al.*, 2014). This approach will allow for the construction of the vertical component of covariance, which is needed in order to understand how PBLH can be used to correct atmospheric profiles through the use of profile and PBLH statistics. We use profiles from nearby model grid points and have tested the system with varying numbers of grid points in the ensemble. An ensemble Kalman filter would likely give different covariance information, but the basic relationship between the state variable profiles and the PBLH are determined by the model in the same manner here.

The NU-WRF simulations, taken from existing forecast runs used for the PECAN campaign (Santanello *et al.*, 2019) are initialized using a National Center for Environmental Prediction (NCEP) Global Forecast System (GFS) reanalysis. The two NU-WRF simulations use the Mellor–Yamada–Janjic (MYJ)[Mellor and Yamada, 1974, 1982; Janjic, 2002] and Mellor-Yamada-Nakanishi-Niino level 2.5 (MYNN) [Nakanishi and Niino, 2009] which are local 1.5 and 2.5 order turbulence closure schemes respectively. The PBLH in each of these models is estimated using the turbulent kinetic energy (TKE) method. The NU-WRF forecast state variables are temperature (T), specific humidity (Q) and velocity (U,V), and we define the forecast vector $\mathbf{x}^f = [T^f \ Q^f \ U^f \ V^f \ (PBLH)^f \ ]$, where we have combined PBLH with the state variables to enable the covariance calculation between them. The vector $\mathbf{x}$ is a column vector, so that the error covariance defined below only includes vertical covariances. The forecast runs are initiated from the NOAA global forecast system (GFS) reanalysis interpolated to the local domain of 30-48N and 84-110 W, with 220×220 lat/lon and 54 vertical levels, at 0 UTC. At this time, the initial state has assimilated all of the convential and satellite observations globally. The two WRF forecast experiments start at 0 UTC, and are run for 22 and 23 hours for the MYJ and MYNN experiments, respectively. We use an ensemble of the 20×20 nearest gridpoints, so that all of the ensemble members are within about 30 km of the lidar observations (since the grid spacing is about 3 km). Generally, larger ensembles using gridpoints farther away will result in larger forecast error covariance because the geographic variability. So this ensemble size was chosen as a balance between ensemble size and geographic localization. The forecast standard deviation for PBLH on the chosen ensem-

ble was around 27 m at 22 UTC. Lidar PBLH observations were made every 25 minutes
on that day in Greensburg, KS (37.6 N, 99.3 W), while balloon soundings were launched
from that location 6 times as part of the Plains Elevated Convection At Night (PECAN;
Gerts et al. 2017).

176       For an EnKF the generalized analysis equations are:

$$\mathbf{x}^a = \mathbf{x}^f + \mathbf{K}(\mathbf{y}^o - H(\mathbf{x}^f)) \tag{1}$$

where $\mathbf{x}^a$ is the analysis state, $\mathbf{x}^f$ is the forecast state, $\mathbf{y}^o$ is the observation vector and
$H$ is the non-linear observation operator. The gain matrix, $\mathbf{K}$ is defined by:

$$\mathbf{K} = \mathbf{P}^f \mathbf{H}^T (\mathbf{H}\mathbf{P}^f \mathbf{H}^T + (\mathbf{R})^{-1}, \tag{2}$$

and $\mathbf{P}^f$ is the forecast error covariance, $\mathbf{R}$ is the observation error covariance and $\mathbf{H}$ is
the linearized observation operator. The matrices $\mathbf{P}^f \mathbf{H}^T$ and $\mathbf{H}\mathbf{P}^f \mathbf{H}^T$ are formed from
the ensemble of forecasts. In the present work, we use the EnOI method, and assimilate
observations one at a time using the the ensemble of profiles described above. In this case,
$\mathbf{x}^a$ and $\mathbf{x}^f$ depend only only vertical level, and $\mathbf{y}^o = y^o$, $\mathbf{R} = (\sigma^o)^2$ and $\mathbf{H}\mathbf{P}^f \mathbf{H}^T =$
$(\sigma^f)^2$ become scaler quantities. The analysis equations are then

$$\mathbf{x}^a = \mathbf{x}^f + \mathbf{K}(y^o - H(\mathbf{x}^f)) \tag{3}$$

and

$$\mathbf{K} = \mathbf{P}^f \mathbf{H}^T ((\sigma^f)^2 + (\sigma^o)^2)^{-1}, \tag{4}$$

The observation error standard deviation supplied by the lidar retrieval is $\sigma^o$, which is
determined from the combined uncertainty of the vertical velocity variance, velocity gra-
dient and backscatter gradient. Generally, when these quantities change rapidly at the
top of the PBL, then the estimated error is small. The error estimates are larger when
(during the evening), the gradients are much more gradual. $\mathbf{H}$ is the linearized obser-
vation operator for PBLH. Because the PBLH is related to the state variables via the
two PBL physics schemes, determining $\mathbf{H}$ would require linearizing the PBL physics at
every analysis time. Rather, here we use the EnOI described above to get:

$$\mathbf{P}^f \mathbf{H}^T \approx \left\langle (\mathbf{x}^f - \mu_{\mathbf{x}}{}^f) \, (H(\mathbf{x}^f - \mu_{\mathbf{x}}{}^f))^T \right\rangle \tag{5}$$

and

$$\mathbf{H}\mathbf{P}^f \mathbf{H}^T = (\sigma^f)^2 \approx \left\langle H(\mathbf{x}^f - \mu_{\mathbf{x}}{}^f) \, (H(\mathbf{x}^f - \mu_{\mathbf{x}}{}^f))^T \right\rangle \tag{6}$$

where $\mu_{\mathbf{x}}{}^f$ is the mean forecast state of the ensemble of profiles. See Houtekamer and Zhang (2016) for a review of ensemble Kalman filter techniques.

We expect the correlation between the airmass within the PBL and the free troposphere to drop away rapidly, because of limited intereactions between them. We found that this can cause errors in the analysis profiles if error covariance between the state variables and PBLH is allowed to continue into the troposphere. To reduce these errors we have added an exponential decay starting at the model level closest to the PBLH ($k_{PBLH}$) to define a vertical localization factor:

$$C_{loc} = exp\left[-\alpha\left(\frac{k - k_{PBLH}}{k_{PBLH}}\right)^2\right] \tag{7}$$

where $k$ is the model level and $\alpha = 8$ is an experimentally determined factor. The factor $C_{loc}$ is multiplied by the vertical covariance in (5) to ensure that the covariance between the PBLH and the state variables becomes small within a couple of model levels into the free troposphere.

Equations 3-4 are solved at each hour using the nearest lidar profile observation in time, and the resulting analysis fields are compared to radiosonde profiles when the latter are also available. There are 22 or 23 analyses (for each forecast run), and 6 times where comparison with radiosonde profiles are made. We focus on the impact of the assimilation on the state variables T, Q, U and V rather than the PBLH because only the state variables would be retained by a forecast.

## 3 Results

This section describes the NU-WRF simulation results, the assimilation of PBLH into these forecasts, and the relationship between the assimilation impact and the time varying forecast and observation error covariances.

### 3.1 NU-WRF simulations

The one day NU-WRF simulations are presented in this section. Figure 1 shows the PBLH during that day, derived from the two NU-WRF forecasts, lidar observations and soundings. We have determined the sounding PBLH using the parcel method (Holzworth, 1964), which defines the top as the height where the potential temperature first exceeds the ground temperature. The lidar PBLH (black *, derived using the method

reported in Bonin, 2018) closely matches the radiosonde estimates (green triangles) in the late evening to nighttime (2-7 UTC), while it is somewhat lower late afternoon to early evening (18-24 UTC). The two NU-WRF forecasts differ from the observations depending on the time of day. During nighttime and early morning the MYJ (red triangles) and MYNN (blue squares) forecasts are higher than the observations, then rise less than the lidar observations in the late morning and early afternoon (12-17 UTC, there are no radiosonde measurements to compare to here) before rising much higher than the observations in the late afternoon (18-24 UTC).

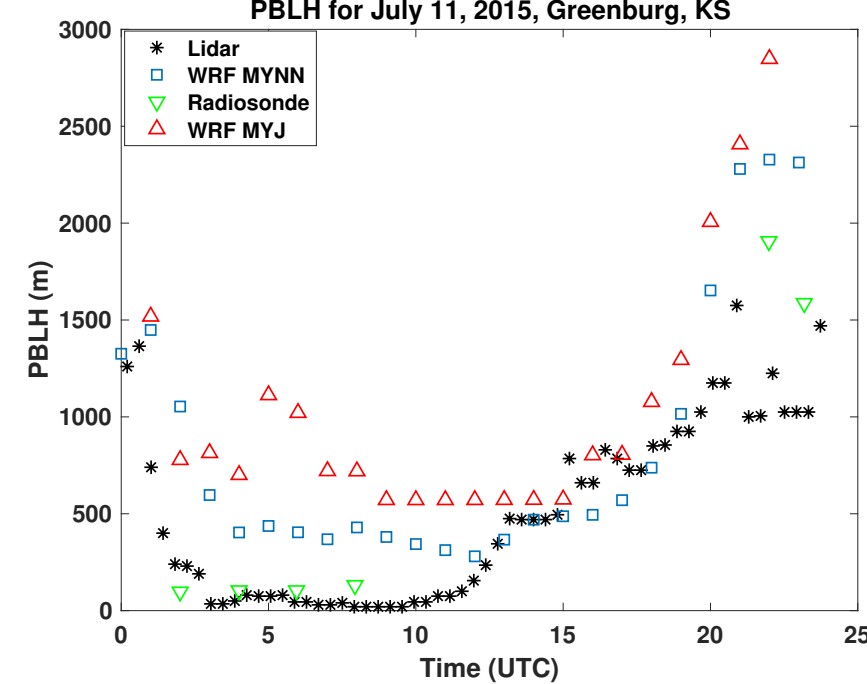

c

**Figure 1.** PBLH vs UTC time for July 11, 2015 for lidar backscatter (black *), WRF model - MYJ (red triangles), WRF model - MYNN (blue squares), and radiosonde observations using parcel method (green triangles).

### 3.2 Impact of assimilation on state variables

Since we are primarily interested in the impact of the assimilation on state variables within the boundary layer, in Figures 2 and 3 we plot the RMS difference between the model and the independent (unassimilated) radiosonde profiles from the surface to roughly the top of the boundary layer in the late afternoon. This corresponds to the first

8 layers, or about 800 mb. We use a fixed number of layers so as to make the compar-
isons of the RMS differences consistent during the day, rather than computing the RMS
over a different number of layers as the PBL grows during the day. For the temperature
forecast, the RMS difference would is

$$RMS(t_a) = \left[ \frac{1}{8} \sum_{i=1}^{8} (T_i^f - T_i^{sonde})^2 \right]^{1/2} \tag{8}$$

where $t_a$ is the analysis time and "i" represents the model level. Figures 2 and 3 show
the RMS differences with the radiosonde profiles throughout the day for the forecasts
(blue x) and analyses (red squares) for potential temperature (a), water vapor mixing
ratio WV (b) and the U (c) and V (d) components of velocity.
During the night (2-9 UTC), the assimilation has a relatively smaller impact on
the potential temperature RMS differences (upper left) in the early morning (6 and 8
UTC), and the two forecasts have similar accuracy. By late afternoon (22 and 23 UTC,
note that the MYJ forecast stops at 22 UTC) the radiosonde comparisons show that the
assimilation reduces RMS differences in the potential temperatures by around $1.5K$ for
MYJ and $2K$ for MYNN. The water vapor mixing ratio (upper right) also has little im-
pact from the assimilation between 2 and 8 UTC, but at 22 UTC (the next radiosonde
profile) the RMS differences for both MYJ and MYNN analyses increase by at least $1.5\times$
$10^{-3} kg/kg$ in the late afternoon. The U-velocity profiles (lower right) show small dif-
ferences between the MYJ and MYNN through 8 UTC (3 a.m. local time) and the as-
similation increases the RMS differences with radiosonde profiles by nearly $1m/s$ start-
ing at 22 UTC for both models. The V-velocity profiles (d) begin to differ between MYJ
and MYNN for the forecasts at 8 UTC ($0.5m/s$ decrease), and assimilation also decreases
the RMS differences with radiosondes in late afternoon by $1.5 - 2m/s$.
We would like to understand why there is a smaller impact during night time and
early morning, whereas there are decreases in the RMS differences in temperature and
V velocity and increases in moisture and U velocity in the late afternoon. To this end,
we plot the forecast, analysis and radiosonde profiles (T, Q, U and V) at 4 UTC (11 p.m.
local time) and 22 UTC (5 p.m. local time) in Figures 4-7. At 4 UTC, (Figures 4,5) these
clearly indicate that there are small corrections made by the assimilation, as the red and

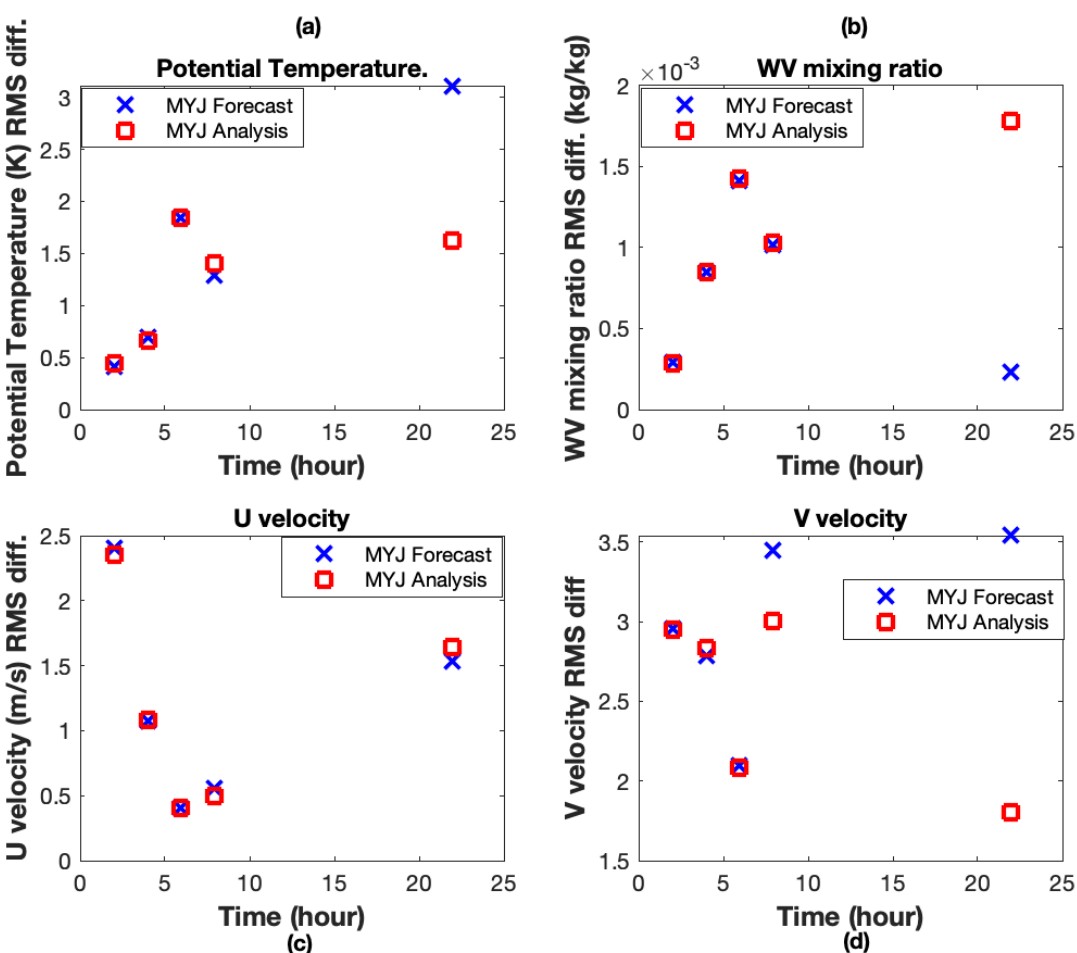

**Figure 2.** RMS difference for lowest 8 layers, vs. time of forecast (blue x) and analysis (red square) with radiosonde profiles for potential temperature (a), water vapor (b), U velocity (c) and V velocity (d).

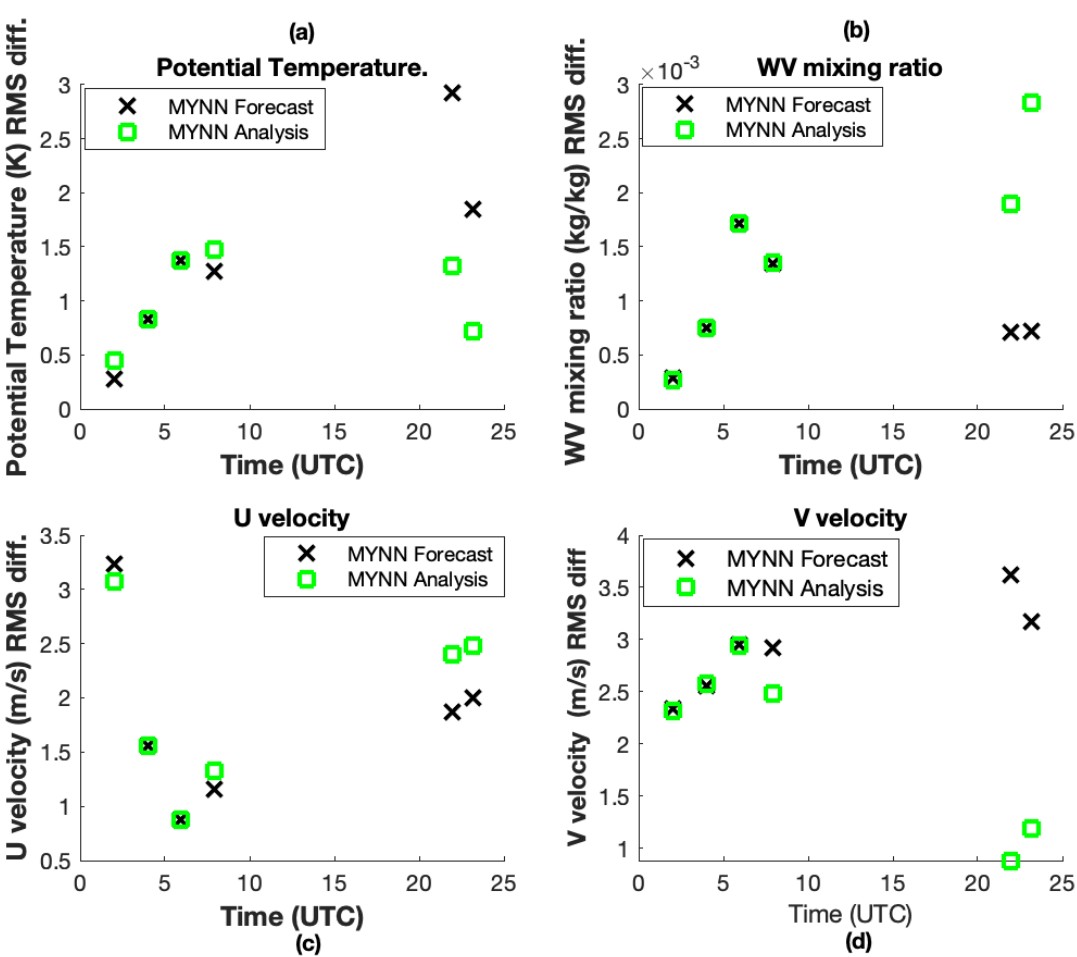

**Figure 3.** Same as Figure 2, but for MYNN PBL model, with forecast (black x) and analysis (blue square).

blue profiles closely overlap. But it also shows that the profiles (particularly temperature and moisture) more accurately follow the radiosonde profiles (except for the U velocity above the PBL), meaning that that any substantial corrections would have made the profiles worse relative the the radiosonde profiles and ultimately degrade the next PBLH forecast. In contrast, Figure (1) shows that the forecast PBLH (particularly MYJ) is quite a bit higher than the lidar observation at 4 UTC. In the late afternoon Figures 6 and 7 indicate that there are large differences between the forecasts and radiosonde profiles for all of the state variables. The forecast PBLH values differ substantially from the lidar measurements as well. The correction to the forecast profiles generally pushes the analyses towards the independent radiosonde profiles, particularly for temperature and V velocity. So the forecasts that predicted both PBLH and state variables with relatively greater accuracy in the early morning were not corrected, while the less accurate afternoon forecast was drawn towards the independent radiosonde measurements. The assimilation also made changes to the vertical velocity (W) in the afternoon, but there is no independent data to compare with so we have not included it.

The WV is shown to be increased by the assimilation (since WV and PBLH are negatively correlated and higher PBLH corresponds to lower WV levels in the PBL models), but the analysis overshoots the radiosonde WV profile for MYNN, hence causing the increase in the water vapor RMS difference in Figures 2 and 3. The MYJ forecast for WV is mostly too high, so the analysis also increases the RMS difference. Compared to temperature, WV is highly variable in time and space and it has been shown in the past that slanted balloon trajectories underestimate the WV present (Demoz et al 2006; Crook, 1996). The U velocity difference with the radiosonde is larger for the analysis, but this correction is more difficult because the differences (at least for MYJ) are both positive and negative and the PBLH observation only contains a single piece of information. The V velocity is, on the other hand, greatly improved by the assimilation. These analysis profiles show that, for this one analysis time, the assimilation is pushing the state variables in the proper direction for temperature, V velocity and moisture, though the moisture correction overshoots the readiosonde profile. PBLH is not a prognostic variable, so that the analysis PBLH values are not retained and therefore cannot directly affect the next forecast. But it is important to note that the temperature and moisture profiles are changed by the assimilation in a way that indicates that the next forecast is likely to have a more accurate PBLH estimate. Figures 6 and 7 both show that the

<sup>297</sup> level at which the potential temperature begins to rise and the WV mixing ratio begins

<sup>298</sup> to drop has been moved to a level much closer to that observed by the lidar. We do not

<sup>299</sup> make forecasts from the analysis fields, but these profiles show promise for improved PBLH

<sup>300</sup> forecasts when cycling experiments are done in a future implementation.

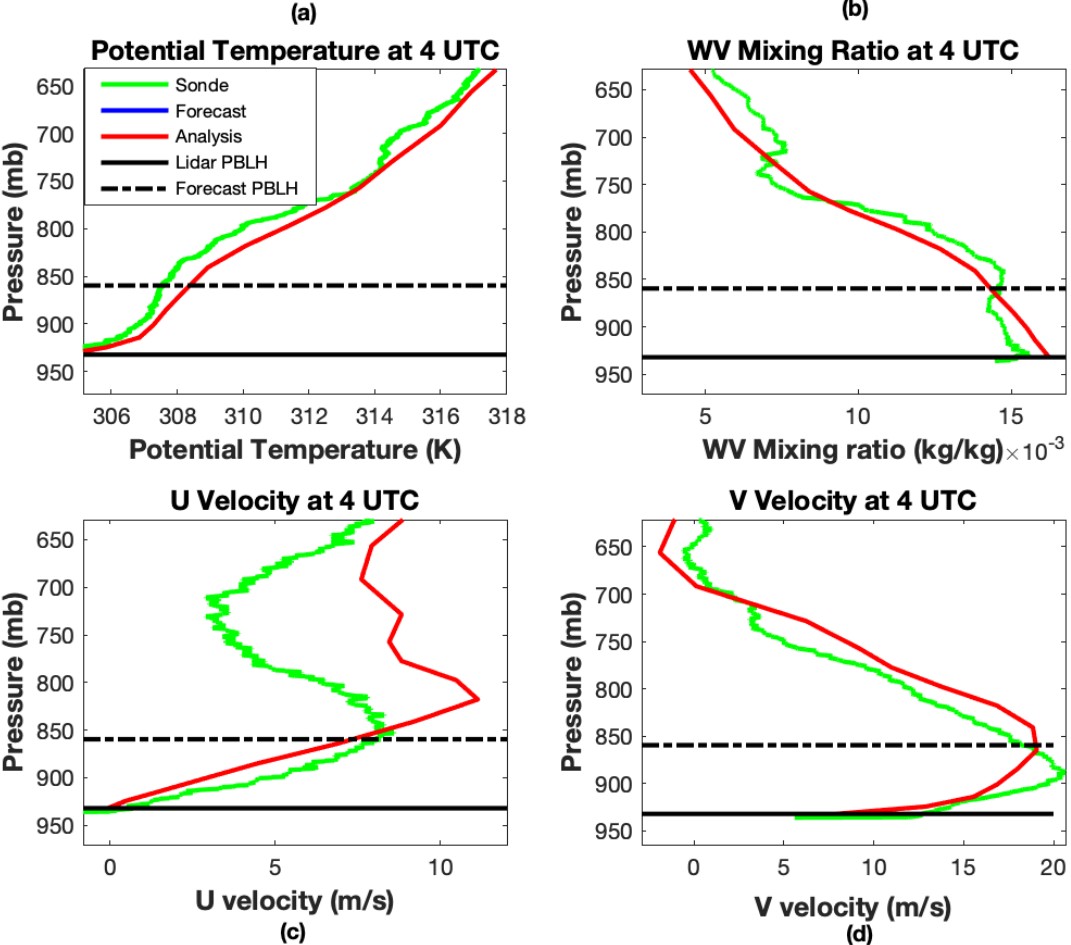

**Figure 4.** Profiles from radiosonde (green), forecast (blue) and analysis (red) for potential temperature (a), water vapor mixing ratio (b), u-velocity (c) and v-velocity (d) at 4 UTC, July 11, 2015 in Greensburg, KS. The model uses the MYJ physics parameterization.

<sup>301</sup> ### 3.3 Ensemble error covariances

<sup>302</sup> The increasing differences between PBLH and profile forecasts from early morn-

<sup>303</sup> ing to late afternoon only partly explain the much larger impact of the assimilation at

<sup>304</sup> 22 UTC. We can also analyze the assimilation by investigating the error covariance be-

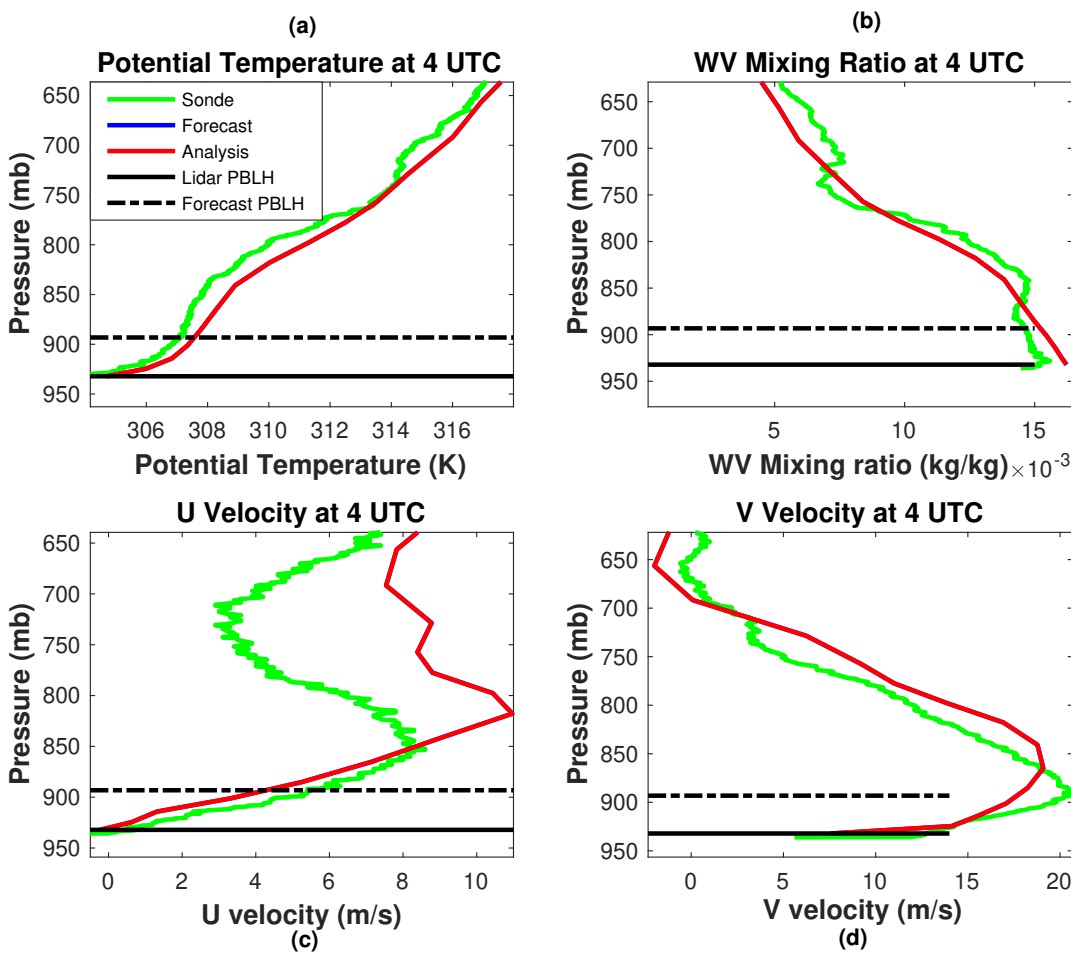

**Figure 5.** Same as figure 4 except using MYNN model.

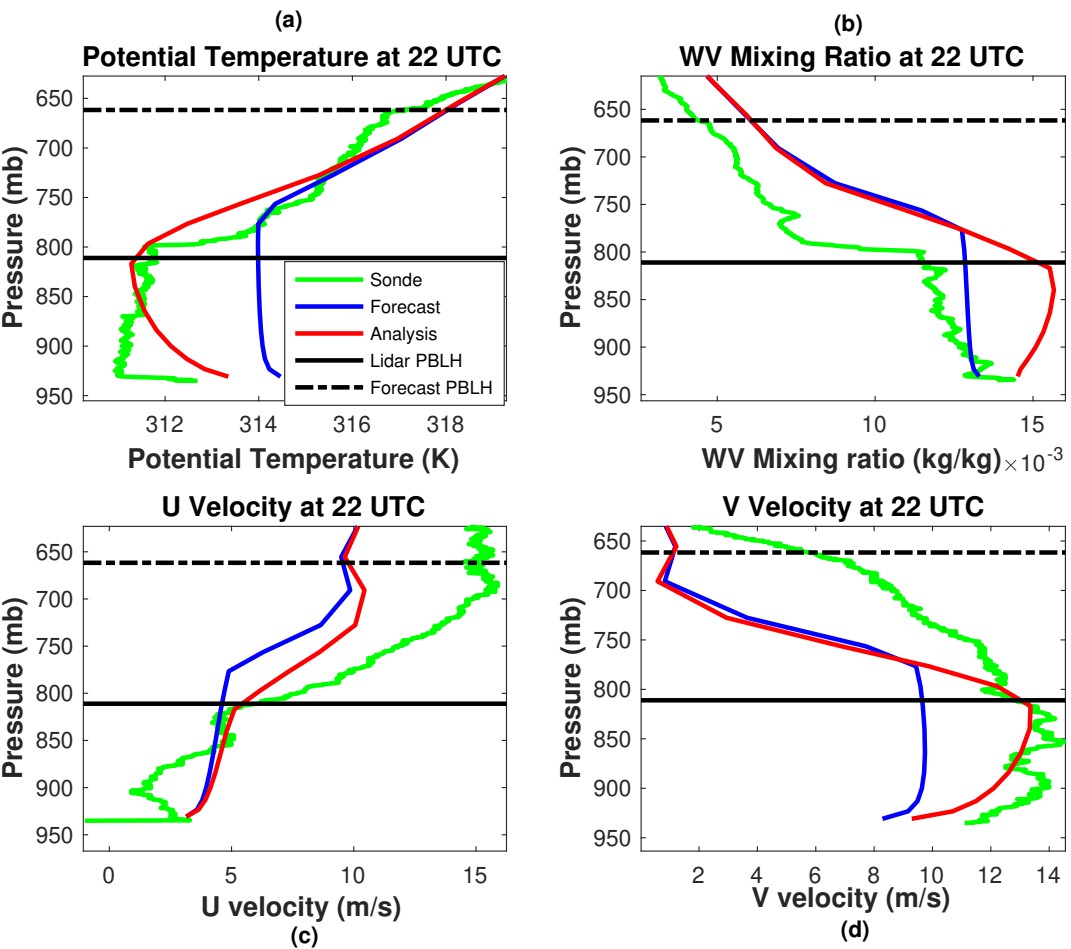

**Figure 6.** Same as figure 4 except using except at time 22 UTC.

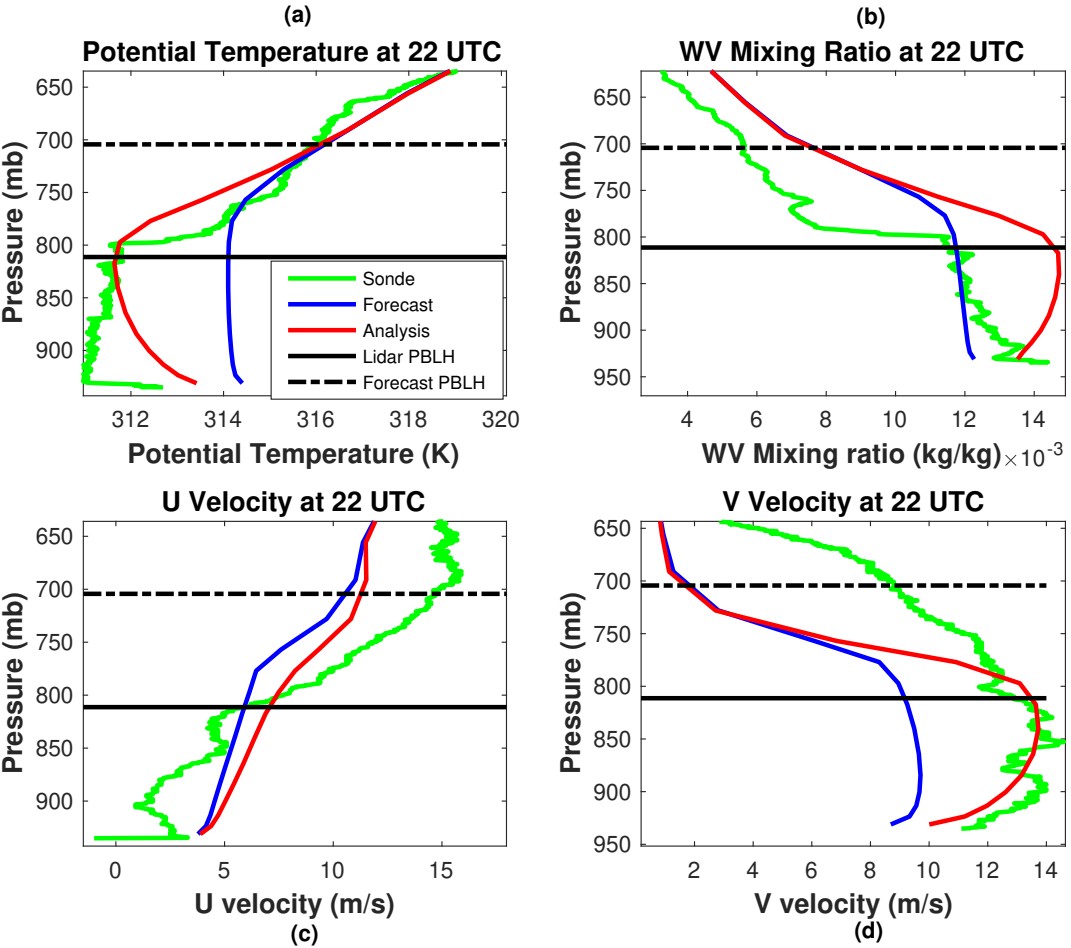

**Figure 7.** Same as figure 6 except using MYNN model.

tween PBLH and each of the state variables ($\mathbf{P}^f\mathbf{H}^T$) and the relative error variances in
observation space ($\mathbf{HP}^f\mathbf{H}^T$ and $\mathbf{R}$). We show $\mathbf{P}^f\mathbf{H}^T$ in Figure 8 for the MYNN PBL
physics model at the 6 radiosonde times. The covariance with temperature is always pos-
itive, and grows by a factor of 4 by late afternoon near the surface. The covariance with
WV is mostly negative and grows by roughly a factor of 5, while the covariance with the
two components of velocity oscillate between positive and negative and shows less con-
sistent growth. Thus, the largest impact of assimilation on temperature and moisture
occurs in late afternoon while more limited velocity corrections are largely constrained
by the correlations determined by the ensemble of model forecast states. In addition, the
covariance between PBLH and the U velocity are substantially smaller than those with
the V velocity. This means that spurious correlations between PBLH and U might be
present, given the relatively small ensemble and the geographic variation of the ensem-
ble members. The error variances are also plotted at the radiosonde times in Figure 9,
which shows that the observation errors are much larger than the forecast errors dur-
ing evening and early morning times (2,4,6,8 UTC) and then become relatively smaller
in the late afternoon (22,23 UTC). This is an additional contributing factor to the min-
imal impact of PBLH observations early in the day and the much larger impact in the
afternoon.

## 4 Discussion and Conclusions

These offline data assimilation experiments indicate that assimilating ground based
lidar backscatter and wind measurements of PBLH into a regional NWP model will likely
lead to corrections to profiles within the PBL, particularly when, in the future, this ap-
proach is applied to an EnKF assimilation system with cycling. Using two NU-WRF fore-
casts over a period of one day with different PBL physics models, we show how the state
variables, T, WV, U and V can be corrected using an assimilation system with ensem-
ble based error covariances. During the night and early morning the assimilation has rel-
atively little impact on the state variables, but by late afternoon the temperature field
is drawn closer to independent radiosonde measurements. We have shown that the lack
of data impact early in the day is the due to the relatively higher accuracy of the model
and lack of correlation between the forecast PBLH and temperature profiles at that time.
Later in the day, when the model is less accurate in predicting the growth of the bound-
ary layer, the data begins to draw the analyses mostly toward the independent radiosonde

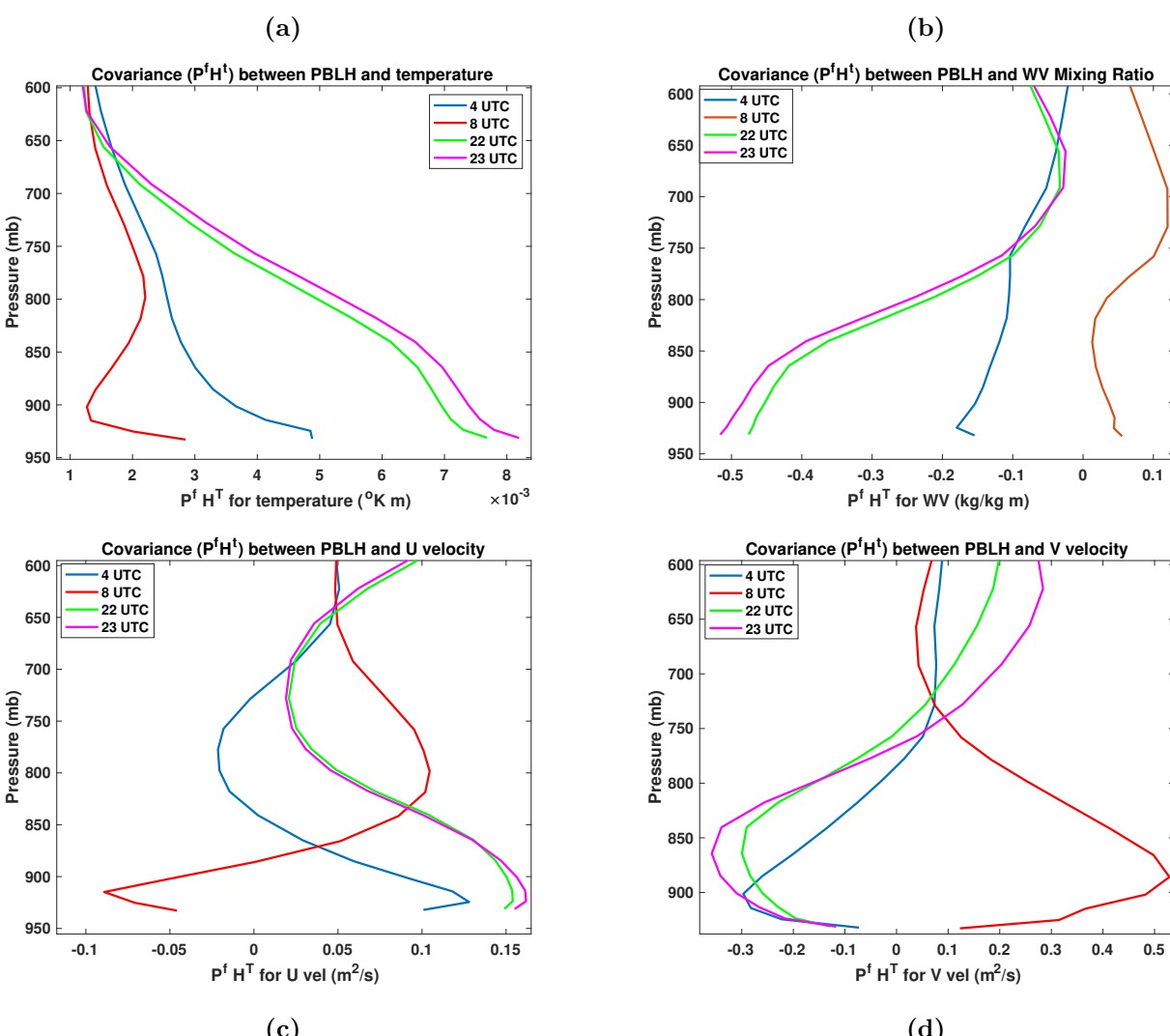

**Figure 8.** Covariance $\mathbf{P}^f\mathbf{H}^T$ between PBLH and temperature (a), water vapor (b), U velocity (c) and V velocity (d), at times 4, 8, 22 and 23 UTC, for PBL physics model MYHH.

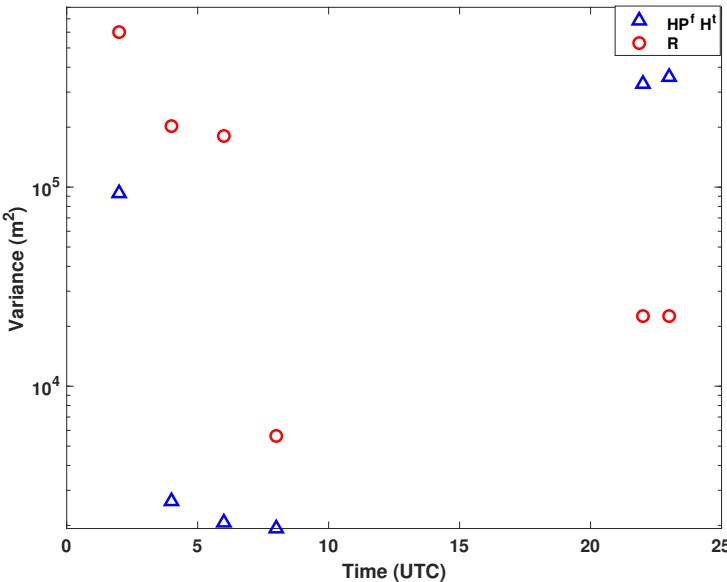

**Figure 9.** Forecast ($\mathbf{HP}^f\mathbf{H}^t$) and observation ($\mathbf{R}$) error covariance for the PBL physics model MYHH at the 6 radiosonde times.

profiles. The assimilation overcorrected the water vapor mixing ratio in the direction of radiosonde data, and this could likely be tuned in an assimilation system. And it corrected the the V velocity component by a smaller amount, and reduced differences with the radiosonde profiles for the V velocity. These corrections are the result of ensemble computed error covariances between the PBLH and the state variable profiles within the PBL. The results here indicate that this approach has some potential to be used in a forecast system in a way that that the PBLH observational information could be carried forward in time so as to impact the forecast accuracy within the PBL. An additional value of assimilating PBLH is its close connection with the PBL scheme used in the model. The ensemble covariances between PBLH and the different state variables are controlled through the PBL physics scheme. This has an impact on the corrections made to the profiles within the PBL, which can be used as another way to evaluate the physics parameterizations. For example, the MYJ and MYNN result in forecast profiles that differ, particularly in WV in the late afternoon. And the differences in reponse to assimilation are an indication of how the two different PBL schemes affect the covariance between PBLH and the state variables. However, a full evaluation would require that the assimilation be implemented into a cycling data assimilation system.

This work is intended only to demonstrate a necessary first step in terms of how ensemble statistics can help to constrain profiles within the PBL by assimilating PBLH observations. A more complete demonstration of this approach will require the construction of an EnKF, which should be run over many days with a variety of weather patterns, including significantly warmer(cooler) and wetter(drier) days. This is needed to show how the assimilated PBLH observations will impact future forecasts within the PBL. More of the PBL physics schemes need to be investigated as well, since the correlation between PBLH and state variables will vary widely depending on which scheme is used. An estimate of the forward operator error should be included in the algorithm as well. There are also differences in the way PBLH is computed in the PBL physics schemes, and the methods used for radiosonde observations (see Hegarty, et al., 2018). This will impact the manner in which the assimilation and resulting forecasts are validated. The larger uncertainty in the lidar PBLH retrievals during nighttime (Figure 9) mean that the assimilation will not significantly constrain the model state within the PBL during this period. So it would be very helpful to complement PBLH observations with thermodynamic and kinematic profile observations, partuculary overnight. The fact that PBLH is a non-negative variable means that the O-F statistics will likely be non-Gaussion so that the assimilation algorithm would need to include an extension to handle this possibility (e.g. Cohn, 1997).

In addition, a cycling EnKF will involve spatial covariances in both horizontal and vertical directions, and will allow for both inflation and horizontal localization. This will enable further tuning of the system to optimize the analysis state relative to the independent radiosonde observations. The PBLH assimilation withn the EnKF framework could be done in any of numerous existing EnKF assimilation systems that connect with WRF, including NU-WRF (Lidard-Peters *et al.*, 2015) and WRF-DART (Anderson *et al.*, 2009). Future development of PBLH assimilation algorithms will also need to address the effect of the different definitions of PBLH, such as the TKE method used the physics schemes and the backscatter and wind profile method used in the retrievals.

## 5 Acknowledgments

A. Tangborn was funded through the JCET cooperative agreement with NASA Goddard Space Flight Center. B. Demoz was funded by National Science Foundation award (AGS-1503563) to the University of Maryland, Baltimore County and through NOAA

Cooperative Science Center in Atmospheric Sciences and Meteorology, funded by the Educational Partnership Program at NOAA in collaboration with Howard University. J. Santanello was funded through a NASA Decadal Survey Study Team grant.

The careful reading and comments by Rohith Muraleedharan Thundathil and the three anonymous reviewers has helped to greatly improve the quality of this paper.

## 6 Data Sets

PECAN (`https://data.eol.ucar.edu/master_list/?project=PECAN\verb`) data are archived by NCAR/EOL, which is funded by NSF. The forecast and analysis fields produced for this work are stored at https://alg.umbc.edu/pecan/.

## 7 Competing Interests

The authors declare that they have no conflict of interest.

## 8 Author Contributions

Andrew Tangborn built the assimilation system, with input from Jeffrey Anderson on the algorithm. Belay Demoz and Brian Carroll provided the lidar observations. Joseph Santanello provided background information on PBL physics. All of the authors contributed to writing and revising the paper.

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
