# Peer review of "Assimilation of lidar planetary boundary layer height observations."

_Atmospheric Measurement Techniques, 2020_

## Referee Comment (RC1) · Anonymous Referee #1 · 8 Aug 2020

This paper provides a nice proof-of-concept study to show the impact of assimilating boundary layer height observations collected by Doppler lidars. When assimilating these data using a system similar to a Kalman filter, the authors' show improved fit between NU-WRF analyses and collocated rawinsonde observations. The impacts are largest when assimilated during the late afternoon and smallest at night. The author's attribute this temporal shift in the assimilation impacts due to the model background correctly predicting the PBLH overnight (zero innovation). These results are also shown for two different PBL parameterization schemes, though the author's rarely draw any conclusions regarding their comparative performance.

Overall, I found the scientific goals of this paper interesting and something that AMT readers would be interested in. However, I have a number of issues with the inter-

pretation of the results. I am confused by the appearance of exactly zero impact to the evening/morning profiles given a non-zero innovation and non-zero covariances. I also do not agree with the presentation of time- series to show the assimilation impacts when there is such a large gap in data between 08-22 UTC. Finally, I request a large number of clarifications regarding the presentation of both the methods and the figures, as well as additional references for various statements. Given these comments, I recommend major revisions.

Major comments:

1. L216-L225 and the assimilation impact from 00-08 UTC: I am confused why there appears to be exactly zero assimilation impact during the overnight hours. In Fig. 2, the forecast and analysis lines are exactly on top of each other for the first four verification times. However, looking at Figure 1, the innovation (observation minus background) can still be large during the overnight hours, especially for MYNN. For example, Fig. 1 shows MYNN underpredicting the PBLH by ∼300 m at 0400 UTC. Given that the error covariances at this time are also non-zero, as shown in Figure 7, I would expect at least some impact. Additionally, Fig. 2 shows the MYJ and MYNN RMS values of (T, Q, U, V) being exactly equal. This seems odd given that Fig. 1 shows them predicting very difference values of PBLH. Thus, I wonder if there is some error in the assimilation scheme or analysis techniques that could be leading to this appearance of zero-impact.

2. Figure 2: Given the large data gap between 08-22 UTC and the use of only six soundings for verification, I disagree with the use of a time-series to show the assimilation impacts. This choice leads to the appearance of the impact linearly increasing between 08-22 UTC, when it likely shows a very different shape in reality. Additionally, statements line L207 (water vapor mixing ratio has little impact until 22 UTC) are not correct given that there is likely an impact beginning at 12 UTC when the innovation becomes much larger. It is just that you do not have any radiosondes confirm that. I suggest removing this figure, or at least removing the lines that connect the verification times. I also suggest removing any text referring to temporal changes in the impacts

3. Figures 3-6: These figures can be difficult to interpret given the lack of any innovation information. I found myself having to flip back and forth between these plots and Figure 1 to try and understand why the impacts were small at certain times. Please include the forecast PBLH on these figures, or at least annotate the innovation (Lidar PBLH minus forecast PBLH).

4. Overly general writing: Sometimes I felt that the author's made general statements when those statements only were instead meant to refer to a specific PBL scheme. For example, it is stated in the abstract that assimilating PBLH observation improves water vapor relative to independent radiosondes. However, this does not appear to be the case for MYJ (figure 2). Additional examples of this are at L217, L228, L241, and L279. Please check and modify such statements throughout the manuscript.

Minor comments:

L16 (and throughout): the use of "sonde" instead of "rawinsonde" or "radiosonde" feels a little informal. Please correct.

L46: I suggest stating "non-local flux schemes" since that helps separate those types of schemes from the local TKE schemes.

L50: The sentence beginning "These varying and distinct" is confusing. I suggest rewording.

L58: I am not sure what the point of this reference to GPSRO is. This seems oddly specific and overly verbose. It could probably be removed.

L73: Jumping from the discussion of ceilometers to lidars feels a little abrupt. Please improve the flow between these two paragraphs (i.e., stating something like "we use Doppler lidars as a proxy to determine the impact of assimilating PBLH from a network of ceilometers").

L74: Please provide a little more information on the brand and type of Doppler lidar used. There were multiple instruments employed during PECAN so it wouldn't hurt to

be more specific.

L81-L82 and L94-100: I suggest moving some of this content into the methodology sections. It doesn't really fit in an introduction.

L82: I would like to see more details on how PBLH is estimated from the Doppler lidar data instead of just giving the reference. This could provide needed context for understanding how different the estimates of PBLH are between the lidars, radiosondes, and the PBL parameterization schemes.

Introduction: One thing I was curious about when reading this manuscript is the motivation for assimilating PBLH instead of directly assimilating the wind profiles collected by the lidars. Lidar wind profiles have been assimilated in the past with positive results shown (Kawabata et al. 2014, Degelia et al. 2020), so why go through the extra steps of deriving PBLH from those data? I suggest adding a sentence or two in the introduction to discuss this.

L116 and EnOI discussion: It seems that the EnOI computes the covariance structure with a spatial component (covariance over a given area). How representative is that of the EnKF method which can estimate covariance at a single point? Does that cause any issues with extrapolating these impacts to a hypothetical EnKF system (i.e., L269)?

L127: Is the same method used to compute PBLH for both the stable and convective boundary layer? I know MYNN is supposed to be more accurate at night compared to MYJ.

L132: Please also list the grid-spacing for these simulations.

L111: Is there a reference for the NU-WRF forecasts run during PECAN?

L137-139: Is this true? I would expect that the covariance/correlation would be smaller when computed over a larger region?

L148: Please include more information on the observation error variance! This term

is equally as important in the analysis as the background error covariance. How is it determined? How do you convert the lidar wind errors into PBLH errors? Do you include any representation factors?

L151-155: It might be good to reference an EnKF paper for these approximations since it is the same technique applied here (i.e., Houtekamer and Zhang 2016).

L162: Please state the chosen value of $\alpha$.

L172-L179: Much of this paragraph detailing the model configuration is repeated from the methods paragraph beginning at L124. Please reduce.

L181: Is there a reference for the parcel method?

L196: Why 800 hPa? Why not compute the RMS from the surface to the top of the PBL since you know its height? Please provide some justification for this number.

Figure 1: Should there be an additional sounding observation during the late evening? I only see five green triangles, but you reference six radiosonde launches. Additionally, there are six verification points shown in the time series plots.

L207: I recommend using absolute differences instead of percent changes.

L213: I disagree with saying the assimilation reduces by the RMS "significantly". Is statistical significance computed here? Also, this sentence appears to be referring to the impact to U-wind in Fig. 2c, of which the impacts look extremely small to me.

Figs. 2-6: Please add (a,b,c,d) headings to each figure to match the figure caption.

Figure 2: I recommend changing (hour) in the x-axis to (UTC) to be consistent with the text. Also please be consistent between saying "U wind" and "zonal velocity" in the figure captions.

L221-223, L276: I disagree with the statement of the model profiles "accurately" following the radiosonde profiles in Figs. 3-4. For example, the u-wind shows errors of

~4 m/s, and the mixing ratio errors can be as large as 1-2 g/kg which is not exactly "accurate".

Figs. 3-6: I recommend reducing the vertical extent of these profiles you are primarily focusing on impacts within the PBL. Maybe 800 hPa since that is what you use for the RMS calculations?). Also, I notice that some of the axis labels and formats are different between these figures, so please be consistent.

L235-L38: I am not sure that the discussion of vertical localization fits with the rest of this paragraph.

L244: I do not understand this statement that suggests PBLH is more representative of water vapor flux. Please elaborate.

L279-282. There is a mix-up of tenses here. The first sentence uses present tense (the water vapor mixing ratio is over corrected), while the second sentence uses past tense (the assimilation corrected...). Please fix. I also noticed other instances of this so I recommend doing a pass to fix issues throughout the manuscript.

Typos and wording changes

1. L5-6: Please spell out the affiliations.

2. L35-39: this sentence is overly long. Please split up or condense.

3. L42: Add a comma after "Alternatively".

4. L55: Please use UTC instead of "Z" time to be consistent with the rest of the paper.

5. L62: Change the reference to Hicks et al. 2016 to use parenthesis instead of brackets.

6. L114: The sentence beginning "Instead we use..." seems broken. Please fix.

7. L198: ntop is not used in this equation. Please remove.

8. L233: Fix the spelling for "independent".

[Figure]

9. L238: Please define "WV".

10. L267: Please change "assimilation" to "assimilating".

11. L288: Sentence beginning "The covariances" is broken. Please fix.

―――――――――――――――――――――

---

## Referee Comment (RC2) · Anonymous Referee #2 · 13 Aug 2020

Review of "Assimilation of lidar planetary boundary layer height observations"

This manuscript describes a study of assimilating lidar PBLH data into 22 hourly forecasts from NU-WRF model on July 11, 2015 in Greensburg, Kansas. The tests are performed in a stand-alone approach, where the analysis results don't feed back to the forecast. The use of ensemble background error covariance for multi-variable relationships is commonly used in the ensemble and 4DEnVar data assimilation systems. This study is a good example to demonstrate the advantage of this approach. The results show that the PBLH data have little impact during the early morning but improve the temperature and velocity in the later afternoon. The manuscript is well written, and test results are clearly presented.

Having said that, I have some comments listed below:

1. The definition of PBLH. As described on lines 77-82, for PBLH data calculation, the Doppler shift of the backscattered signal is used to calculate wind speed as a function of range, which can then be used to produce a multitude of wind and turbulence variables useful for PBL characterization (e.g. vertical velocity variance and signal-to-noise ratio variance). The PBLH algorithm applied for this study combines several such aerosol and wind variables for PBLH measurement and was described at length in Bonin et al. (2018). The PBLH in the model is estimated using the total kinetic energy (TKE) method. The two definitions are different but seem close enough. Is there a way to show to what extent the two PBLH definitions are comparable?

2. The vertical localization factor. How is the parameter alpha in equation (6) chosen? According to the equation, this parameter works the same way for layers both above and below the PBL height, for example, if k_PBLH=4, then C_loc at layer 3 is the same as C_loc at layer 5. However, that seems not the case in Fig. 5.

3. Equation (7). Where is number "8" coming from? The top of boundary layer is not a constant during the 22 hours, which can be seen clearly in Figures 3-6.

4. In the abstract, it states that water vapor is improved by assimilating lidar PBLH. However, Fig. 5 shows that it is degraded.

---

## Referee Comment (RC3) · Anonymous Referee #3 · 26 Aug 2020

General comments:

The paper describes the impact of assimilating PBLH observations from active remote-sensing Doppler lidar system into the NU-WRF model. The assimilation resulted in reducing the PBLH RMS difference compared to independent radiosonde measurements in late afternoon although the impact was little during the night and early morning. Also the forecast covariance of state parameters with the PBLH variable have also been analysed. I think the article presents a novel research and would be a nice contribution supporting active remote-sensing network data assimilation for mesoscale as well as synoptic systems. However, I would request the author to make a few changes in the manuscript before publishing.

Major comments:

[Figure]

1. Line 212 – "….the assimilation reduces the RMS differences with sonde profiles significantly by 22 UTC for both models." From Fig. 2, the RMS difference of potential temperature, WVMR and V component of velocity have reasonable impact but there is little or no impact on U wind. Please correct the statement if it was a mistake, or, if not, please elaborate how the impact is significant. Also please adjust the Y axis limits of V wind to the same as that of U wind.

2. In Figs. 3 and 4 both analysis and forecasts profiles of potential temperature, WVMR and velocities, U and V, coincide each other at 4 UTC. However, in Fig. 1, the PBLH at 4 UTC is not the same for MYNN forecast although MYJ forecast PBLH has the same value as the radiosonde. The PBLH difference of MYNN forecast to radiosonde is around 300 m from Fig. 1 which creates a doubt regarding Fig. 4 (MYNN scheme) at least if not Fig. 3. May be the innovation was not large enough to create an impact in the assimilation system. Also another reason for doubt is due to the significant magnitude of covariance of PBLH with the variables for 4 and 8 UTC. Hence, I would suggest the author to create the same Figs. 3 and 4 with an additional background profile (may be use a dashed line of the same colour) for each of the variables to remove the doubt.

Minor Comments:

1. I would suggest the author to include a brief description of Doppler lidar just after the ceilometers. A brief description on the pros and cons of Doppler lidar (with references to the system used) and how it is superior to ceilometers could be added.

2. Line 134 - Please add some more details regarding the assimilation design in the methodology section. The sentence "…experiments are all less than 24 hours from the most recent global analysis" is not clear enough for readers. Line 98 - "The assimilation is done on 22 hourly WRF forecast fields…" may be omitted or modified after the above addition in the methodology section.

3. Line 178 – Radiosonde launches were 6 times in total. The reader understands

MYJ has 5 radiosonde comparisons since it stopped at 22 UTC whereas MYNN has 6 radiosondes. Please clarify this point.

Typos and corrections:

1. Line 59 – "Wulfmeyer et al. 2015" not found in the reference section.

2. Line 67 - Please check "Brooks, 2003". I could not find the reference in the reference section.

3. Line 144 – The sentence "Instead we use. . .error statistics" should be corrected.

4. Line 119 – "We use profiles from. . ." feels like repetition from line 115.

5. Line 129 – Please describe "W".

6. Line 220 – Please change "plue" to "blue".

7. Line 244 – "Demoz et al 2006; Crook, 1996" could not be found in the reference section.

8. Line 272 – "an" is used twice, please correct.

9. The following references were found in the reference section without citation in the manuscript. Please cite these wherever necessary.

"Banks, R. F., J. Tiana-Alsina, F. Rocadenbosch, and J. M. Baldasano (2015) Performance evaluation of the boundary-layer height from lidar and the Weather Research and Forecasting Model at an urban coastal site in the north-east Iberian Peninsula. Bound.-Layer Meteor., 157, 265–292, https://doi.org/ 10.1007/s10546-015-0056-2."

"Cohen, A.E., S.M. Cavallo, M.C. Coniglio and H.E. Brook (2015), A Review of Planetary Boundary Layer Parameterization Schemes and Their Sensitivity in Simulating Southeastern U.S. Cold Season Severe Weather Environments, Wea. Forecat., 30, 591-612."

"Tucker, S.C., S.J. Senff, A.M. Weickmann, W.A. Brewer, R.M. Banta, S.P. Sandberg,

D.C. Law and R.M. Hardesty (2009), Doppler Lidar Estimation of Mixing Height Using Turbulence, Shear, and Aerosol Profiles, J. Atmos. Ocean Tech., 26, 673-688."

---

## Author Comment (AC1) · 15 Sep 2020

**Assimilation of lidar planetary boundary layer height observations**

Andrew Tangborn, Belay Demoz, Brian Carroll, Joseph Santanello and Jeffrey Anderson

**Response to reviewer 1**

**Reviewer 1**

Major comments:

*1. L216-L225 and the assimilation impact from 00-08 UTC: I am confused why there appears to be exactly zero assimilation impact during the overnight hours. In Fig. 2, the forecast and analysis lines are exactly on top of each other for the first four verification times. However, looking at Figure 1, the innovation (observation minus background) can still be large during the overnight hours, especially for MYNN. For example, Fig. 1 shows MYNN underpredicting the PBLH by 300 m at 0400 UTC. Given that the error covariances at this time are also non-zero, as shown in Figure 7, I would expect at least some impact. Additionally, Fig. 2 shows the MYJ and MYNN RMS values of (T, Q, U, V) being exactly equal. This seems odd given that Fig. 1 shows them predicting very difference values of PBLH. Thus, I wonder if there is some error in the assimilation scheme or analysis techniques that could be leading to this appearance of zero-impact.*

The analysis increments are never zero, but are much smaller from 0-8 UTC. But we also found an inconsistancy in the definitions geopotential height and PBLH (the former defined above ground level and the latter above sea level), and have redone the assimilation. You can now see somewhat larger changes in some of the profiles during this time. Also, the smaller increments are also due to the variation in the lidar observation error estimates, which vary substantially during the day.

*2. Figure 2: Given the large data gap between 08-22 UTC and the use of only six soundings for verification, I disagree with the use of a time-series to show the assimilation impacts. This choice leads to the appearance of the impact linearly increasing between 08-22 UTC, when it likely shows a very different shape in reality. Additionally, statements line L207 (water vapor mixing ratio has little impact until 22 UTC) are not correct given that there is likely an impact beginning at 12 UTC when the innovation becomes much larger. It is just that you do not have any radiosondes confirm that. I suggest removing this figure, or at least removing the lines that connect the verification times. I also suggest removing any text referring to temporal changes in the impacts C2*

We have removed the lines between the radiosonde measurement times. It became difficult to see both of the PBL model forecasts without them, so we split Figure 2 into two figures (2 and 3). The text has been changed to reflect this.

*3. Figures 3-6: These figures can be difficult to interpret given the lack of any innovation information. I found myself having to flip back and forth between these plots and Figure 1 to try and understand why the impacts were small at certain times. Please include the forecast PBLH on these figures, or at least annotate the innovation (Lidar PBLH minus forecast PBLH).*

We have included the forecast PBLH in the profile plots.

*4. Overly general writing: Sometimes I felt that the author's made general statements when those statements only were instead meant to refer to a specific PBL scheme. For example, it is stated in the abstract that assimilating PBLH observation improves water vapor relative to independent radiosondes. However, this does not appear to be the case for MYJ (figure 2). Additional examples of this are at L217, L228, L241, and L279. Please check and modify such statements throughout the manuscript.*

The text has been changed to reflect the changes to assimilation (described in item 1 above), and to make the comments more specific. Though the L228 comment was concerning a plot that we didn't include. And L279 is a more speculative statement on how changing the state variables would be carried forward in time, though we have modified this to make it more qualified.

Minor comments:

*L16 (and throughout): the use of "sonde" instead of "rawinsonde" or "radiosonde" feels a little informal. Please correct.*

This has been changed.

*L46: I suggest stating "non-local flux schemes" since that helps separate those types of schemes from the local TKE schemes.*

done.

*L50: The sentence beginning "These varying and distinct" is confusing. I suggest rewording.*

It has been rewritten as: "The variety of definitions PBLH make it difficult to effectively evaluate existing models or develop new ones."

*L58: I am not sure what the point of this reference to GPSRO is. This seems oddly specific and overly verbose. It could probably be removed.*

Removed.

*L73: Jumping from the discussion of ceilometers to lidars feels a little abrupt. Please improve the flow between these two paragraphs (i.e., stating something like "we use Doppler lidars as a proxy to determine the impact of assimilating PBLH from a network of ceilometers").*

We have added further wording to make this transition less abrupt.

*L74: Please provide a little more information on the brand and type of Doppler lidar used. There were multiple instruments employed during PECAN so it wouldn't hurt to be more specific.*

This information has been added.

*L81-L82 and L94-100: I suggest moving some of this content into the methodology sections. It doesn't really fit in an introduction.*

We don't agree with making this move. These details are not about the assimilation algorithm, which is described in the methodology section. I don't think details about the observations belongs in methodology because the retrievals are not a part of the methodology developed in this work. Further, you are asking for more details about the lidar observations in this section already (L74 and L82). So it seems best to leave this the way it is.

*L82: I would like to see more details on how PBLH is estimated from the Doppler lidar data instead of just giving the reference. This could provide needed context for understanding how different the estimates of PBLH are between the lidars, radiosondes, and the PBL parameterization schemes.*

Additional description of the PBLH retrieval algorithm has been added.

*Introduction: One thing I was curious about when reading this manuscript is the motivation for assimilating PBLH instead of directly assimilating the wind profiles collected by the lidars. Lidar wind profiles have been assimilated in the past with positive results shown (Kawabata et al. 2014, Degelia et al. 2020), so why go through the extra steps of deriving PBLH from those data? I suggest adding a sentence or two in the introduction to discuss this.*

We have added "But we are interested assimilating the PBLH observations directly because the ceilometer network described above will focus on these retrievals, and satellite missions which measure PBLH are also planned.".

*L116 and EnOI discussion: It seems that the EnOI computes the covariance structure with a spatial component (covariance over a given area). How representative is that of the EnKF method which can estimate covariance at a single point? Does that cause any issues with extrapolating these impacts to a hypothetical EnKF system (i.e., L269)?*

We are only using the EnOI to compute covariance in the vertical direction, since we are concerned with the profile correction. With the EnKF one would also compute the horizonal structure as well. In addition, the variance estimate will dependend on the distance spacing of the profiles, with a larger distance resulting in a larger variance. We chose a relatively small set of 20x20 grid locations to minimize this effect. In the EnKF, one would also include inflation and horizontal localization. These would need to be worked out when an EnKF is constructed for this data type. We have added a couple of sentences on this.

*L127: Is the same method used to compute PBLH for both the stable and convective boundary layer? I know MYNN is supposed to be more accurate at night compared to MYJ.*

The PBLH estimate approaches are the same at all times. The values changed in this version as we found a inconsistency in the definition of PBLH, so now the MYNN scheme is more accurate during the night. We think the manuscript is reasonably clear on this.

*L132: Please also list the grid-spacing for these simulations.*

The grid spacing is 3km, and was already in the first version of the manuscript.

*L111: Is there a reference for the NU-WRF forecasts run during PECAN?*

The only reference at this point is Santanello, et al. 2019, which is an AGU meeting abstract.

*L137-139: Is this true? I would expect that the covariance/correlation would be smaller when computed over a larger region?*

Over a large region the meteorological conditions become more varied, so the variance becomes larger.

*L148: Please include more information on the observation error variance! This term is equally as important in the analysis as the background error covariance. How is it determined? How do you convert the lidar wind errors into PBLH errors? Do you include any representation factors?*

The PBLH observations are determined from the combined velocity variance dropoff, wind speed gradient and backscatter dropoff. The uncertainty is smallest where these values decay rapidly over a short distance. When the dropoff is more gradual (as in the morning), the estimated uncertainty is much larger. This is described in the text.

*L151-155: It might be good to reference an EnKF paper for these approximations since it is the same technique applied here (i.e., Houtekamer and Zhang 2016).*

Reference added.

*L162: Please state the chosen value of $\alpha$.*

$\alpha = 8$ has been added to the paper.

*L172-L179: Much of this paragraph detailing the model configuration is repeated from the methods paragraph beginning at L124. Please reduce.*

We reduced the deails in the results section slightly.

*L181: Is there a reference for the parcel method?*

We added Holzworth, 1964.

*L196: Why 800 hPa? Why not compute the RMS from the surface to the top of the PBL since you know its height? Please provide some justification for this number.*

We chose $800mb$ because this is roughly the maximum height of the PBL on this day. If we chose to compute the RMS up to levels that vary with the PBLH, then it would be difficult to make direct comparisons between the RMS at different times in the day. We have added a sentence on this in the paper.

*Figure 1: Should there be an additional sounding observation during the late evening? I only see five green triangles, but you reference six radiosonde launches. Additionally, there are six verification points shown in the time series plots.*

The sixth radiosonde PBLH has been added to the figure.

*L207: I recommend using absolute differences instead of percent changes.*

We have changed this to absolute differences.

*L213: I disagree with saying the assimilation reduces by the RMS "significantly". Is statistical significance computed here? Also, this sentence appears to be referring to the impact to U-wind in Fig. 2c, of which the impacts look extremely small to me.*

We have changed the discussion here, and removed the term "significantly".

*Figs. 2-6: Please add (a,b,c,d) headings to each figure to match the figure caption.*

We changed the caption to read upper left, upper right, etc instead of using letters to identify the figure location.

*Figure 2: I recommend changing (hour) in the x-axis to (UTC) to be consistent with the text. Also please be consistent between saying "U wind" and "zonal velocity" in the figure captions.*

These have all been changed to "U wind".

*L221-223, L276: I disagree with the statement of the model profiles "accurately" following the radiosonde profiles in Figs. 3-4. For example, the u-wind shows errors of 4 m/s, and the mixing ratio errors can be as large as 1-2 g/kg which is not exactly "accurate".*

This has been changed so that the profiles are described as "more accurate" duing the early morning than they are in the late afternoon.

*Figs. 3-6: I recommend reducing the vertical extent of these profiles you are primarily focusing on impacts within the PBL. Maybe 800 hPa since that is what you use for the RMS calculations?). Also, I notice that some of the axis labels and formats are different between these figures, so please be consistent.*

We have kept the upper limit at about 600 mb because we felt it was important to show how the profile reveals the location of the top of the PBL (either the model or observation estimates), so it was helpful to include a region above the PBL. This also enables us to show how much correction is made above the PBLH. But we have made the fonts on the axis labels consistent.

*L235-L38: I am not sure that the discussion of vertical localization fits with the rest of this paragraph.*

We removed these two sentences. This had already been discussed in the methodology section.

*L244: I do not understand this statement that suggests PBLH is more representative of water vapor flux. Please elaborate.*

We removed the last two sentences from this discussion.

*L279-282. There is a mix-up of tenses here. The first sentence uses present tense (the water vapor mixing ratio is over corrected), while the second sentence uses past tense (the assimilation corrected: : :). Please fix. I also noticed other instances of this so I recommend doing a pass to fix issues throughout the manuscript.*

These sentences have been corrected.

Typos and wording changes

*1. L5-6: Please spell out the affiliations.*

Done.

*2. L35-39: this sentence is overly long. Please split up or condense.*

Done.

*3. L42: Add a comma after "Alternatively".*

Done.

*4. L55: Please use UTC instead of "Z" time to be consistent with the rest of the paper.*

Done.

*5. L62: Change the reference to Hicks et al. 2016 to use parenthesis instead of brackets.*

Done.

*6. L114: The sentence beginning "Instead we use: : :" seems broken. Please fix.*

Done.

*7. L198: ntop is not used in this equation. Please remove.*

Changed to $i = 8$.

*8. L233: Fix the spelling for "independent".*

Done.

*9. L238: Please define "WV".*

Done.

*10. L267: Please change "assimilation" to "assimilating".*

Done.

*11. L288: Sentence beginning "The covariances" is broken. Please fix.*

Done.

---

## Author Comment (AC2) · 15 Sep 2020

**Assimilation of lidar planetary boundary layer height observations**

Andrew Tangborn, Belay Demoz, Brian Carroll, Joseph Santanello and Jeffrey Anderson

**Response to reviewer 2**

*1. The definition of PBLH. As described on lines 77-82, for PBLH data calculation, the Doppler shift of the backscattered signal is used to calculate wind speed as a function of range, which can then be used to produce a multitude of wind and turbulence variables useful for PBL characterization (e.g. vertical velocity variance and signal-to-noise ratio variance). The PBLH algorithm applied for this study combines several such aerosol and wind variables for PBLH measurement and was described at length in Bonin et al. (2018). The PBLH in the model is estimated using the total kinetic energy (TKE) method. The two definitions are different but seem close enough. Is there a way to show to what extent the two PBLH definitions are comparable?*

This Doppler lidar was not making measurements capable of direct TKE retrieval, only TKE proxies (such as vertical velocity variance), so an explicit apples-to-apples comparison is not possible here. To make further inference would be speculative, so instead we only present and discuss this best possible PBLH measurement from the Doppler lidar to assess model performance.

Once a much larger number of PBLH lidar observations are obtained, along with radiosonde observations, it would be worthwhile to generate some statistics on this, on both bias and random differences. We have 6 sonde observations to compare with our forecasts, and with these we can show here is how the lidar observations can impact the thermodynamic profiles within the PBL using assimilation of the lidar observations. With a better understanding of differences between the two PBLH schemes, and a much larger data set to compare with, it's likely that further improvements can be made.

*2. The vertical localization factor. How is the parameter alpha in equation (6) chosen? According to the equation, this parameter works the same way for layers both above and below the PBL height, for example, if $k_P BLH = 4$, then $C_l oc$ at layer 3 is the same as $C_l oc$ at layer 5. However, that seems not the case in Fig. 5.*

We have redone the assimilation to fix a couple of inconsistancies in the code, including this. The profile plots now show the vertical localization above and below the top of the PBL, though the final form of any localization that would be needed will be more clear once this is implemented with an enKF.

*3. Equation (7). Where is number "8" coming from? The top of boundary layer is not a constant during the 22 hours, which can be seen clearly in Figures 3-6.*

The maximum extent of the PBL in the late afternoon is at layer 8, and we felt is was more consistent to compare the same levels at each time, rather than comparing a much smaller number of layers during the night and early morning. This is explained further in the text.

*4. In the abstract, it states that water vapor is improved by assimilating lidar PBLH. However, Fig. 5 shows that it is degraded.*

We have corrected this statement. A more accurate statement is that the assimilation changes the water vapor profile in the right direction, but the increment is too large, so that the RMS difference with the radiosondes increases. This would require additional tuning in an EnKF.

---

## Author Comment (AC3) · 15 Sep 2020

**Assimilation of lidar planetary boundary layer height observations**

Andrew Tangborn, Belay Demoz, Brian Carroll, Joseph Santanello and Jeffrey Anderson

**Response to reviewer 3**

*1. Line 212 – ": : :.the assimilation reduces the RMS differences with sonde profiles significantly by 22 UTC for both models." From Fig. 2, the RMS difference of potential temperature, WVMR and V component of velocity have reasonable impact but there is little or no impact on U wind. Please correct the statement if it was a mistake, or, if not, please elaborate how the impact is significant. Also please adjust the Y axis limits of V wind to the same as that of U wind.*

We have made some changes here due to changes in the solution in this revision. Please see the response to the last item from reviewer 2. We have made the y-axis limits the same for the U and V plots.

*2. In Figs. 3 and 4 both analysis and forecasts profiles of potential temperature, WVMR and velocities, U and V, coincide each other at 4 UTC. However, in Fig. 1, the PBLH at 4 UTC is not the same for MYNN forecast although MYJ forecast PBLH has the same value as the radiosonde. The PBLH difference of MYNN forecast to radiosonde is around 300 m from Fig. 1 which creates a doubt regarding Fig. 4 (MYNN scheme) at least if not Fig. 3. May be the innovation was not large enough to create an impact in the assimilation system. Also another reason for doubt is due to the significant magnitude of covariance of PBLH with the variables for 4 and 8 UTC. Hence, I would suggest the author to create the same Figs. 3 and 4 with an additional background profile (may be use a dashed line of the same colour) for each of the variables to remove the doubt.*

We have made some corrections to the code, which has changed both the innovations and the corrections to the profiles. We have also put the lidar observation levels (in pressure) on the profile plots to make more lear the magnitude of the innovations. There are now some what larger corrections to the profiles in the early morning, and none of them is zero.

Minor Comments:

*1. I would suggest the author to include a brief description of Doppler lidar just after the ceilometers. A brief description on the pros and cons of Doppler lidar (with references to the system used) and how it is superior to ceilometers could be added.*

We have added further details on the Doppler lidar.

*2. Line 134 - Please add some more details regarding the assimilation design in the methodology section. The sentence ": : :experiments are all less than 24 hours from the most recent global analysis" is not clear enough for readers. Line 98 - "The assimilation is done on 22 hourly WRF forecast fields: : :" may be omitted or modified after the above addition in the methodology section.*

We have added further explanation as to where the forecasts start (0 UTC) from the NOAA global forecast system (GFS) with a final initialization at 0 UTC.

*3. Line 178 – Radiosonde launches were 6 times in total. The reader understands MYJ has 5 radiosonde comparisons since it stopped at 22 UTC whereas MYNN has 6 radiosondes. Please clarify this point.*

The missing radiosonde has been added to the PBLH plot.

Typos and corrections:

*1. Line 59 – "Wulfmeyer et al. 2015" not found in the reference section.*

Added.

*2. Line 67 - Please check "Brooks, 2003". I could not find the reference in the reference section.*

Added.

*3. Line 144 – The sentence "Instead we use: : :error statistics" should be corrected.*

We have changed this sentence, but we think you had meant line 115.

*4. Line 119 – "We use profiles from: : :" feels like repetition from line 115.*

We think you meant line 144 here. And we have shortened and simplified the sentence to avoid repetition.

*5. Line 129 – Please describe "W".*

W is the vertical velocity, but we are not showing it here because there are not observations to validate it. So we have removed it.

*6. Line 220 – Please change "plue" to "blue".*

Done.

*7. Line 244 – "Demoz et al 2006; Crook, 1996" could not be found in the reference section.*

These citations have been added to the reference list.

*8. Line 272 – "an" is used twice, please correct.*

Done.

*9. The following references were found in the reference section without citation in the manuscript. Please cite these wherever necessary.*

*"Banks, R. F., J. Tiana-Alsina, F. Rocadenbosch, and J. M. Baldasano (2015) Performance evaluation of the boundary-layer height from lidar and the Weather Research and Forecasting Model at an urban coastal site in the north-east Iberian Peninsula. Bound.-Layer Meteor., 157, 265–292, https://doi.org/ 10.1007/s10546-015-0056-2."*

*"Cohen, A.E., S.M. Cavallo, M.C. Coniglio and H.E. Brook (2015), A Review of Planetary Boundary Layer Parameterization Schemes and Their Sensitivity in Simulating Southeastern U.S. Cold Season Severe Weather Environments, Wea. Forecat., 30, 591-612."*

*"Tucker, S.C., S.J. Senff, A.M. Weickmann, W.A. Brewer, R.M. Banta, S.P. Sandberg, D.C. Law and R.M. Hardesty (2009), Doppler Lidar Estimation of Mixing Height Using Turbulence, Shear, and Aerosol Profiles, J. Atmos. Ocean Tech., 26, 673-688."*

These References have been removed.

---

## Referee Report (RR1)

I'd like to thank the authors for answering my questions and addressing my concerns. The manuscript overall is better.

Regarding my previous PBLH definition question, I understand that the authors would like to defer the evaluation/discussion to future work.  Having said that, as it is important to understand the nature of the observation operator and use corresponding model variable for data assimilation, I would suggest the authors to add in the conclusions saying that the observation operator and the model variable "issue" still needs to be addressed.

Minor comments:

Line 324: the sentence "particulary when this approach is applied to an EnKF assimilation system with cycling" should be removed. If I understand correctly, the experiments performed in this work are stand-alone analysis, no forecasts are issued from the analyses, and no cycling is involved.

Figure numbers are shown as ?? in the manuscript.

---

## Referee Report (RR2)

**Article summary**
This manuscript assesses the feasibility of planetary boundary layer height (PBLH) observations to improve the initial conditions of numerical weather prediction (NWP) models. To address this problem, off-line experiments are conducted in which Doppler lidar PBLH retrievals are combined with synthetically generated model profiles from two different simulations. An Ensemble Optimal Interpolation (EnOI) algorithm is used to assimilate the lidar-derived PBLH observations. The state vector is augmented to incorporate the prior PBLH values diagnosed by the underlying PBL scheme. Six independent radiosonde measurements are used to determine the consistency of the updated model profiles. The authors find that more significant corrections to the prior model profiles occur in the late afternoon hours when the background ensemble is (i) less accurate and (ii) contains more significant cross-variable ensemble covariances. Overall, it is also shown that the assimilation of PBLH retrievals leads to mixed results: while there are visible improvements in the analysis profiles of potential temperature and the V-component of the wind, the corrections to the water vapor mixing ratio and the U-component of the wind are not consistent with the verifying radiosonde measurements.

**Decision**: accept with major revisions
This proof-of-concept study provides an important benchmark for the future assimilation of ceilometer-based PBLH retrievals, so I strongly recommend its publication in AMT. I praise the authors for constructing a data assimilation system that is sophisticated enough to address the key objectives in their study, but also sufficiently simple to minimize any unnecessary computational costs. Nevertheless, there are several important aspects that the authors should address to further enhance the impact of their results. These are summarized below alongside with numerous other minor comments and stylistic/formatting suggestions.

**Major comments**

*1. The presentation of the data assimilation results needs to be refined.*

*(a) Interpretation of the observation misfits*
On L245-L248, the authors justify the small analysis increments in the early morning hours by noting the small deviation between the forecasted thermodynamic and kinematic profiles from the corresponding radiosonde measurements. This conclusion in not consistent with the EnOI update because the innovation $d = y^o - h(x^b)$ only refers to the misfit between observed and forecasted (diagnosed) PBLH values. L248-L249 correctly state that there are relatively large misfits between the observed and prior PBLH, but the authors do not emphasize that these differences control the subsequent corrections of the prior T, MV, U and V profiles.

*(b) The role of prior ensemble spread in the model space corrections*
I also think that the authors need to better quantify the origin of the analysis increments in model space. From Eqs. (2) and (3), it should be clear that the magnitude of the model state corrections depends on two distinct factors: (i) the ensemble cross-covariances $P^f H^T$ and (ii) the scaled innovation $\tilde{d} = \left(R + HP^f H^T\right)^{-1} y^o - h(x^b)$. To justify the observed differences between the model state corrections in the early morning and late afternoon hours, the authors focus on the magnitude of the ensemble covariances (Fig. 8), but pay little attention to how the background ensemble spread in observation space ($HP^f H^T$) affects the scaled PBLH misfit $\tilde{d}$. Aside from contributing to a more complete understanding of the model state corrections,

examination of the prior PBLH spread will also shed some light on the underlying forecast uncertainty.

*(c) Discussion of the PBLH corrections*
While I agree with the authors' approach of focusing on the model state increments, a brief discussion of the PBLH corrections is also warranted. Such a discussion is important because the PBLH increments have a direct impact on the subsequent model state corrections (e.g., Anderson 2003). Intuitively, if a data assimilation system is not effective in estimating the directly observed quantities, it will also struggle to constrain the unobserved model state variables. Therefore, I encourage the authors to elaborate on the extent to which the analyzed PBLH values were able to move closer to the independent radiosonde-based PBLH retrievals.

*2. I found the authors' justification on the importance of assimilating PBLH retrievals particularly appealing (L102-L104), but the literature overview on past efforts of assimilating Doppler lidar measurements (L99-L104) needs to be corrected and expanded.*

First, none of the cited studies (Hu et al. 2019; Coniglio et al. 2019; Degelia et al. 2019) report improvements to the PBLH; instead they examine the forecast performance with respect to convective initiation. Second, the aforementioned papers assimilate horizontal wind profiles derived from the VAD technique. This comes in contrast with other studies (e.g., Kamineni et al. 2003) which focus on the impacts from assimilating thermodynamic lidar profiles. To put this study into a broader context, I suggest that the authors place more emphasis on the aforementioned differences and further expand on their literature overview starting on L98. Apart from discussing the past efforts referenced above, the authors should also include some of the most recently published research in this area. For instance, both Degelia et al. (2020) and Chipilski et al. (2020) assimilate Doppler lidar retrievals in the context of nocturnal convective systems. Chipilski et al. (2020) additionally reveal that the assimilation of Doppler lidar wind data affects the characteristics of the stable PBL – a finding which is particularly relevant in the context of the present study.

*3. A critical discussion is also needed to highlight the future challenges of assimilating PBLH retrievals.*

The authors point out that their work is a "necessary first step in terms of how ensemble statistics can help to constrain profiles within the PBL", indicate the need to assimilate PBLH retrievals in real-time EnKF systems under diverse weather regimes, but provide little information on the specific problems of assimilating the PBLH variable. Given that this paper is designed to inform the future incorporation of PBLH retrievals into real-time data assimilation systems, the authors should elaborate on what the most outstanding challenges in this area are. The bullet points below provide a couple of suggestions: while I certainly do not expect the authors to follow all of them, I encourage their incorporation with some of the authors' own concerns.
- The calculation of PBLH is different in model simulations and observations: while the observed PBLH is derived from Doppler lidar measurements of turbulence intensity, horizontal wind profiles and backscatter intensity, the prior PBLH is diagnosed using the specific formulation of a particular PBL scheme. Each of these two methods has its own approximations and, perhaps even more importantly, will not yield the same PBLH value even if the simulated and observed meteorological conditions are identical. If such biases

are not taken into account, the analysis estimate produced by Eq. (3) will be suboptimal. The aforementioned methodological differences also constitute a problem from the viewpoint of forecast verification.

- Treatment of observations errors. On L166, it is stated that the observation error variances are equal to the uncertainty estimates provided by the lidar retrievals. In additional to these measurement errors, operational data assimilation systems typically also consider (i) errors of representation and (ii) errors due to approximations in the observation (forward) operator. These additional error contributions might be important to consider in future efforts. For example, errors of representativity might be especially relevant within the more inhomogeneous stable PBL, whereas the methodological differences in the PBLH computation could be treated, at least to a first degree, as errors in the observation operator.

- The ability of PBLH retrievals to efficiently constrain the model state. Here I offer two different comments. The first one concerns the inability of PBLH retrievals to correct the stable PBL structure despite the large differences in the prior and observed PBLH values. The analysis offered by the authors indicates that the lack of ensemble cross-covariances is a likely explanation. This raises the question, however, whether the small corrections in stable PBLs constitutes a systematic effect induced by the formulation of current PBL schemes. Answering this question is important as one of the focal objectives of the PECAN field campaign was to understand if the information provided by ground-based PBL profilers can improve the traditionally poor forecasts of nocturnal convective systems. The second point the authors might want to consider is how effective the PBLH retrievals would be in the context of other observation networks. A past NRC report (NRC 2009) describes the potential deployment of a nation-wide network of thermodynamic and kinematic PBL profilers, quite similar to the ones employed during the PECAN field campaign. Unlike the ceilometer network that originally motivated this study, the PBL profilers will produce direct observations of model state variables that could make the ceilometer-based PBLH retrievals redundant.

- My last comment is more technical in nature and relates to the theoretical inappropriateness of current data assimilation systems to extract information from PBLH retrievals. By definition, PBLH is a non-negative quantity. As such, it faces the problems similar to those associated with the assimilation of certain moisture variables (e.g., specific humidity; see Dee and da Silva 2003). Because PBLH is a bounded quantity, its distribution will not always be Gaussian. Hence, traditional assumptions used to derive operational data assimilation schemes will be violated (e.g., Bocquet et al. 2010; Bannister et al. 2020). Such deviations from Gaussianity will be particularly visible under stable PBLs (because PBLH is closer to its lower bound of 0m), further complicating the data assimilation problems already noted in this manuscript. It might be possible to remedy the problem of boundedness by adopting certain non-Gaussian extensions of traditional data assimilation algorithms (cf. Cohn 1997; Fletcher and Zupanski 2006; Bishop 2016).

**Minor comments**

1. L32: Momentum is also exchanged in land-atmosphere interactions, please add to the list.

2. L46-L47: "… since aerosols are well mixed throughout the PBL (Hicks et al., 2019)" - this statement is only valid in the context of CBL.

3. L72-L73: "… were not yet available for the campaign we are using". Before providing details on how the Doppler lidar measurements in Greensburg were obtained, the readers would benefit from a short description of the PECAN field campaign as a whole.

4. The use of "PBLH retrievals" is more appropriate than "PBLH measurements" as it emphasizes that PBLH is a derived quantity. There are a couple of instances in the manuscript where this correction needs to be applied.

5. Description of the methods to assimilate the PBLH retrievals (L107-L109). It might be better to replace the statement "either by creating an adjoint of the PBL parameterization scheme" with "either by adopting a variational data assimilation scheme" (or a semantically equivalent expression). Formulating the adjoint of a parameterisation scheme is a specific implementation aspect of variational data assimilation algorithms. If the authors desire so, they could motivate their preference for an EnKF approach in this study by pointing out that ensemble-based methods sidestep the generally difficult task of linearizing the model physics equations.

6. Please cite the original EnKF paper of Evensen (1994) and its subsequent refinement in Burgers et al. (1998) on L123.

7. L124: "… where the analysis state is the estimate with the minimum estimated errors". The original derivation of the Kalman filter minimizes the expected squared errors. Note that the latter corresponds to a minimisation of a $L^2$ error norm rather than the $L^1$ error norm implied on L124.

8. The EnOI was originally introduced by Oke et al. (2002) and also discussed by Evensen (2003), so please make sure to add these references on L133.

9. On L141, the authors mention the names of the two parameterization schemes used in the archived NU-WRF simulations. A brief justification on why these two parameterization schemes were chosen in the original PECAN runs will be helpful.

10. The abbreviation TKE on L145 is commonly used to denote "turbulent kinetic energy" in boundary layer research, so it might be best to consider rewording.

11. State vector definition (L145-L148). Please clarify whether the state variable Q refers to specific humidity, mixing ratio or another moisture variable. The authors should also stress that the state vector **x** in this study represents a vertical column, which is why $P^f$ only refers to the vertical ensemble covariances.

12. Mathematical description of the EnOI algorithm (L160-L191). Here I have a couple of technical remarks aimed at refining the mathematical presentation of the EnOI algorithm.
- Because the definition of the forecasted error covariance in Eq. (1) is too general and not specific to the EnOI algorithm, it might be best to remove it from the discussion.
- Similarly, it is not necessary to write the measurement equation $y^o = \mathbf{H}\mathbf{x}^f$; instead, the authors could simply state that $y^o$ in Eq. (2) represents the PBLH observations retrieved from the Doppler lidar. (As a side comment, the aforementioned measurement equation is only partially complete in the context of filtering theory as it should include a random noise term.)
- It will be best to avoid mixing the vectorial and scalar notations in Eqs. (2) and (3). This could be done by first writing the general form of Eqs. (2)-(3) and then describing how

these were solved by the EnOI algorithm employed in this study. Regarding the latter, the authors should highlight that $y^o$ as well as the corresponding $HP^fH^T$ and $R$ matrices are scalar quantities and that both $P^fH^T$ and $HP^fH^T$ are computed from an ensemble of model profiles, as indicated by Eqs. (4) and (5).

- A brief description is needed to explain how the vertical localization mentioned on L177-L185 was implemented in this study.

13. Opening sentences in Section 3 (L193-L199). Some aspects regarding the description of the NU-WRF simulations were already discussed in the paragraph starting on L141. The missing details found on L193-L199 should be moved back to this paragraph.

14. L205: "… in the late evening to early morning (2-7 UTC)". 7 UTC does not correspond to early morning.

15. L207: "early morning and early afternoon". Please define this period in the same manner as you have done in other places within the text.

16. Instead of using the temperature as an example in Eq. (7), please use a generic variable, say X, to generalize the RMS difference formula. Moreover, instead of explaining the meaning of i=8 on L220, just mention that the index i denotes a model level.

17. Statements regarding the corrections made to the WV profile at 22 UTC (L262, L270-L271 and L302-303). The analyzed WV profile overshoots the observed one only with respect to the MYNN simulation. By contrast, the forecasted values in the MYJ WV profiles are already higher, so increasing the WV values following the DA update acts to further increase the observation misfit. Please make sure to make this distinction while describing your results.

18. Justification regarding the deteriorated estimates of the U profile (L266-L268). The authors correctly acknowledge that the estimation of U is a challenging task when one assimilates integrated quantities like PBLH. However, a similar inference can be made in terms of the V-component profile of the wind and, in fact, further argued that the estimation of V is more challenging due to the presence of sharp gradients in the 750-800 hPa layer (cf. lower-right panels in Figs. 6 and 7). An alternative hypothesis to explain the reported differences in the U and V estimates can be linked to the magnitude of the cross-variable ensemble covariances. The lower two panels of Fig. 8, for example, show that the PBLH-V cross-covariances are much larger than their PBLH-U counterparts. Taking into account that the performance of EnOI (and all other ensemble-based DA methods) is especially susceptible to number of ensemble members, it is quite possible that the small PBLH-U covariances are simply a manifestation of the inherent sampling noise, which would in turn act to degrade the quality of the analyzed U profiles. In theory, the above hypothesis could be tested by comparing results with different ensemble sizes (or with different number of vertical model profiles in the context of this study).

19. L272-L276: The description of how the PBLH retrievals correct the state variables should be either removed or relocated to the methodology section.

20. L284-287: Here the authors state that the "... more limited velocity corrections are largely constrained by the correlations …". This is only partially true as Fig. 8 shows that the V cross-covariances are considerably larger than their U counterparts.

21. L311: Replace "covariances" with "ensemble covariances" to emphasize the underlying computational method.

22. L311: It will be best to change "defined" to "controlled". The ensemble covariances are defined mathematically through the sample covariance formula, but controlled by the characteristics of the T, MV, U and V profiles that enter the PBL parameterization schemes.

23. L314: Did you intend to refer to the "analysis profiles" instead of the "forecast profiles" here?

**Typos and stylistic changes**

1. The authors should consider segmenting their results in Section 3 with a view of enhancing the readability of their manuscript. One possibility would be to divide the results into three subsections. The first one could discuss the discrepancies between modelled and retrieved PBLH values, the second one – how the assimilated PBLH observations correct the observed and unobserved model variables, while the third might focus on interpreting the magnitude of the T, MV, U and V corrections at 04 UTC and 22 UTC.

2. The paragraph starting on L255 could be merged with the preceding one as it provides a summary of the main results.

3. Please correct the spelling of EnKF on L325 and L329.

4. There were several places where "Doppler" was not capitalized.

5. I spotted unintended word repetitions in a couple of places within the text, e.g. L252, L255 and L303.

6. L23: "leading to an increased differences" – remove "an".

7. L36: "… rapidly transported within this layer". It is not clear which layer is being referred to as the previous sentence mentions both the CBL and SBL.

8. L38: "The PBLH is fundamental to …". This sentence provides a very general description on the significance of PBLH and should be mentioned earlier in the paragraph, e.g. after the sentence starting on L32.

9. L42: "penetrates the top" could be changed to "penetrates its top" to make it clear that the authors refer to the PBL top.

10. L186: "This system is solved…". Which system? Please be more specific by listing the relevant equations.

11. L208: "… the MYJ forecasts (red triangles) both are higher than the observations". Please confirm that "both" refers to the MYJ forecasts in the early morning and early afternoon. If this is the case, remove "both" as it is clear from the context.

12. The use of two time periods ["During the night (2-9 UTC) …", "… in the early morning (6 and 8 UTC) …"] makes it hard to interpret the sentence starting on L225.

13. L232: "increase" should be replaced with "increases".

14. L237: "(0.5m/2 decrease)" should be replaced with "(0.5m/s decrease)".

15. L240: "inthe RMS differences" – please separate "in" from "the".

16. L249-L250: "In the late afternoon (Figures 6,7) indicate …" – please remove the brackets and refine the sentence structure.

17. L260: "The WV profile is shown to be increased …". It is the WV values that increase, not the profile itself.

18. L270: Remove "in" from "in show that".

19. L279: "We can also analyze this …". Not clear what "this" refers to, please clarify.

20. L321-L322: "will require the construction of an EnkF, and run over many days" – please correct the wording in this sentence.

**Figures**

1. It would be useful to label the figure panels with (a), (b), etc.

2. Please avoid repetitions in the title and axis labels (e.g., potential temperature in the upper-left panel of Fig. 5).

3. Legends are sitting atop some of the curves in Figs. 4-7. Please make sure that all data is displayed in the revisited figures.

4. Fig. 8: Please include the ensemble covariance units on the x-axis. Please also replace "for PBL physics model MYHH" to "for the MYNN PBL scheme".

**References**

Anderson, J. L., 2003: A local least squares framework for ensemble filtering. *Mon. Wea. Rev.*, **131**, 634–642, https://doi.org/10.1175/1520-0493(2003)131<0634:ALLSFF>2.0.CO;2.

Bannister, R. N., H. G. Chipilski, and O. Martinez-Alvarado, 2020: Techniques and challenges in the assimilation of atmospheric water observations for numerical weather prediction towards convective scales. *Q. J. R. Meteorol. Soc.*, 1–48, https://doi.org/10.1002/qj.3652.

Bishop, C. H., 2016: The GIGG-EnKF: Ensemble Kalman filtering for highly skewed non-negative uncertainty distributions. *Q. J. R. Meteorol. Soc.*, **142**, 1395–1412, https://doi.org/10.1002/qj.2742.

Bocquet, M., C. A. Pires, and L. Wu, 2010: Beyond gaussian statistical modeling in geophysical data assimilation. *Mon. Wea. Rev.*, **138**, 2997–3023, https://doi.org/10.1175/2010MWR3164.1.

Burgers, G., P. J. Van Leeuwen, and G. Evensen, 1998: Analysis scheme in the ensemble Kalman filter. *Mon. Wea. Rev.*, **126**, 1719–1724, https://doi.org/10.1175/1520-0493(1998)126<1719:ASITEK>2.0.CO;2.

Chipilski, H. G., X. Wang, and D. B. Parsons, 2020: Impact of assimilating PECAN profilers on the prediction of bore-driven nocturnal convection: A multiscale forecast evaluation for the 6 July 2015 case study. *Mon. Wea. Rev.*, **148**, 1147–1175, https://doi.org/10.1175/MWR-D-19-0171.1.

Cohn, S., 1997: An Introduction to Estimation Theory. *J. Meteorol. Soc. Japan*, **75**, 257–288, https://doi.org/10.1248/cpb.37.3229.

Coniglio, M. C., G. S. Romine, D. D. Turner, and R. D. Torn, 2019: Impacts of Targeted AERI and Doppler Lidar Wind Retrievals on Short-Term Forecasts of the Initiation and Early Evolution of Thunderstorms. *Mon. Wea. Rev.*, **147**, 1149–1170, https://doi.org/10.1175/MWR-D-18-0351.1.

Dee, D. P., and A. M. da Silva, 2003: The Choice of Variable for Atmospheric Moisture Analysis. *Mon. Wea. Rev.*, **131**, 155–171, https://doi.org/10.1175/1520-0493(2003)131<0155:TCOVFA>2.0.CO;2.

Degelia, S. K., X. Wang, and D. J. Stensrud, 2019: An Evaluation of the Impact of Assimilating AERI Retrievals, Kinematic Profilers, Rawinsondes, and Surface Observations on a Forecast of a Nocturnal Convection Initiation Event during the PECAN Field Campaign. *Mon. Wea. Rev.*, **147**, 2739–2764, https://doi.org/10.1175/mwr-d-18-0423.1.

——, ——, ——, and D. D. Turner, 2020: Systematic evaluation of the impacts of assimilating a network of ground-based remote sensing profilers for forecasts of nocturnal convection initiation during PECAN. *Mon. Wea. Rev.*, **in press**, https://doi.org/https://doi.org/10.1175/MWR-D-20-0118.1.

Evensen, G., 1994: Sequential data assimilation with a nonlinear quasi-geostrophic model using Monte Carlo methods to forecast error statistics. *J. Geophys. Res.*, **99**, https://doi.org/10.1029/94jc00572.

Evensen, G., 2003: The Ensemble Kalman Filter: Theoretical formulation and practical implementation. *Ocean Dyn.*, **53**, 343–367, https://doi.org/10.1007/s10236-003-0036-9.

Fletcher, S. J., and M. Zupanski, 2006: A data assimilation method for log-normally distributed observational errors. *Q. J. R. Meteorol. Soc.*, **132**, 2505–2519.

Hu, J. U. N. J. U. N., N. Yussouf, D. D. Turner, T. A. Jones, and X. U. G. U. A. N. G. Wang, 2019: Impact of ground-based remote sensing boundary layer observations on short-term probabilistic forecasts of a tornadic supercell event. *Wea. Forecasting*, **34**, 1453–1476, https://doi.org/10.1175/WAF-D-18-0200.1.

Kamineni, R., T. N. Krishnamurti, R. A. Ferrare, S. Ismail, and E. V. Browell, 2003: Impact of High Resolution Water Vapor Cross-Sectional Data on Hurricane Forecasting. *Geophys. Res. Lett.*, **30**, 1234, https://doi.org/10.1029/2002GL016741.

NRC, 2009: Observing Weather and Climate from the Ground Up. National Academies Press, 250 pp.

Oke, P. R., J. S. Allen, R. N. Miller, G. D. Egbert, and P. M. Kosro, 2002: Assimilation of surface velocity data into a primitive equation coastal ocean model. *J. Geophys. Res. Ocean.*, **107**, 1–25, https://doi.org/10.1029/2000jc000511.

---

## Author Response (AR2)

**Assimilation of lidar planetary boundary layer height observations**

Andrew Tangborn, Belay Demoz, Brian Carroll, Joseph Santanello and Jeffrey Anderson

**Response to reviewer 2**

*I'd like to thank the authors for answering my questions and addressing my concerns. The manuscript overall is better.*

*Regarding my previous PBLH definition question, I understand that the authors would like to defer the evaluation/discussion to future work. Having said that, as it is important to understand the nature of the observation operator and use corresponding model variable for data assimilation, I would suggest the authors to add in the conclusions saying that the observation operator and the model variable "issue" still needs to be addressed.*

We have added a sentence to this effect.

*Minor comments:*

*Line 324: the sentence "particulary when this approach is applied to an EnKF assimilation system with cycling" should be removed. If I understand correctly, the experiments performed in this work are stand-alone analysis, no forecasts are issued from the analyses, and no cycling is involved.*

This sentence is actually in line 291. And it was meant to describe a hypothetical EnKF. We have added some additional wording to indicate that this is for future efforts to build a cycling EnKF.

*Figure numbers are shown as ?? in the manuscript.*

This is occurs in the version of the paper that shows the changes from the previous manuscript, and since they have a different number of figures, some (or perhaps all) appear as ??. You should be able to see the correct figure numbers in the revised manuscript.

**Response to reviewer 4**

Major comments

*1. The presentation of the data assimilation results needs to be refined.*

*(a) Interpretation of the observation misfits On L245-L248, the authors justify the small analysis increments in the early morning hours by noting the small deviation between the forecasted thermodynamic and kinematic profiles from the corresponding radiosonde measurements. This conclusion in not consistent with the EnOI update because the innovation $d = y - h(x)$ only refers to the misfit between observed and forecasted (diagnosed) PBLH values. L248-L249 correctly state that there are relatively large misfits between the observed and prior PBLH, but the authors do not emphasize that these differences control the subsequent corrections of the prior T, MV, U and V profiles.*

You are correct in that the profiles do not directly impact the analysis increments since PBLH is the only observed quantity assimilated here. And it is also true that the state variables at a given time will impact the PBLH at a later time, and therefore the correction then as well. We have revised these sentences to make it clearer.

*(b) The role of prior ensemble spread in the model space corrections I also think that the authors need to better quantify the origin of the analysis increments in model space. From Eqs. (2) and (3), it should be clear that the magnitude of the model state corrections depends on two distinct factors: (i) the ensemble cross-covariances $P^f H^T$ and (ii) the scaled innovation $d = R + HPfH^T/(y - h(x^f))$. To justify the observed differences between the model state corrections in the early morning and late afternoon hours, the authors focus on the magnitude of the ensemble covariances (Fig. 8), but pay little attention to how the background ensemble spread in observation space ($HP^f H^T$) affects the scaled PBLH misfit $d$. Aside from contributing to a more complete understanding of the model state corrections, examination of the prior PBLH spread will also shed some light on the underlying forecast uncertainty.*

We agree that this is an important comparison to make. We have added a figure that shows both $HP^fH^T$ and $R$ at the 6 times when the sonde data is available, and these indicate that $R$ is much larger during the times when there is little impact from the assimilation.

*(c) Discussion of the PBLH corrections While I agree with the authors' approach of focusing on the model state increments, a brief discussion of the PBLH corrections is also warranted. Such a discussion is important because the PBLH increments have a direct impact on the subsequent model state corrections (e.g., Anderson 2003). Intuitively, if a data assimilation system is not effective in estimating the directly observed quantities, it will also struggle to constrain the unobserved model state variables. Therefore, I encourage the authors to elaborate on the extent to which the analyzed PBLH values were able to move closer to the independent radiosonde-based PBLH retrievals.*

We are assuming that you are not asking if the analysis estimate of PBLH is closer to the observations than the forecast estimate. This would be trivially true, and not important because the analysis PBLH is not a prognostic variable and will have no impact on the next forecast. Instead, we think you mean that are the temperature and moisture analysis fields likely to produce a better PBLH if they were passed through the PBL physics package (which is not done in this work). We can say with some confidence that we think it would improve the PBLH. For example, the height potential temperature profiles can be seen to rise is much closer to the observed PBLH for the analysis. This indicates that the PBL scheme would likely produce a more accurate PBLH from the analysis, and possibly for the next forecast. We have added some comments in the text to explain this.

*2. I found the authors' justification on the importance of assimilating PBLH retrievals particularly appealing (L102-L104), but the literature overview on past efforts of assimilating Doppler lidar measurements (L99-L104) needs to be corrected and expanded. First, none of the cited studies (Hu et al. 2019; Coniglio et al. 2019; Degelia et al. 2019) report improvements to the PBLH; instead they examine the forecast performance with respect to convective initiation. Second, the aforementioned papers assimilate horizontal wind profiles derived from the VAD technique. This comes in contrast with other studies (e.g., Kamineni et al. 2003) which focus on the impacts from assimilating thermodynamic lidar profiles. To put this study into a broader context, I suggest that the authors place more emphasis on the aforementioned differences and further expand on their literature overview starting on L98. Apart from discussing the past efforts referenced above, the authors should also include some of the most recently published research in this area. For instance, both Degelia et al. (2020) and Chipilski et al. (2020) assimilate Doppler lidar retrievals in the context of nocturnal convective systems. Chipilski et al. (2020) additionally reveal that the assimilation of Doppler lidar wind data affects the characteristics of the stable PBL – a finding which is particularly relevant in the context of the present study.*

We have revised the discussion on assimilation of profile observations, including discussion of the Chipilski (2020) abnd Degelia (2020) papers. And we have connected these to works to this paper by adding a couple of sentences on convective initiation withn the PBL.

*3. A critical discussion is also needed to highlight the future challenges of assimilating PBLH retrievals. The authors point out that their work is a "necessary first step in terms of how ensemble statistics can help to constrain profiles within the PBL", indicate the need to assimilate PBLH retrievals in real-time EnKF systems under diverse weather regimes, but provide little information on the specific problems of assimilating the PBLH variable. Given that this paper is designed to inform the future incorporation of PBLH retrievals into real-time data assimilation systems, the authors should elaborate on what the most outstanding challenges in this area are. The bullet points below provide a couple of suggestions: while I certainly do not expect the authors to follow all of them, I encourage their incorporation with some of the authors' own concerns.*

*· The calculation of PBLH is different in model simulations and observations: while the observed PBLH is derived from Doppler lidar measurements of turbulence intensity, horizontal wind profiles and backscatter intensity, the prior PBLH is diagnosed using the specific formulation of a particular PBL scheme. Each of these two methods has its own approximations and, perhaps even more importantly, will not yield the same PBLH value even if the simulated and observed meteorological conditions are identical. If such biases are not taken into account, the analysis estimate produced by Eq. (3) will be suboptimal. The aforementioned methodological differences also constitute a problem from the viewpoint of forecast verification.*

We have added a couple of sentences on the need to study a wide variety of PBL physics schemes with

this assimilation approach. While the different methods to compute PBLH from radiosonde data add to the uncertainty, it's not as important here because we are not trying to improve estimates of the PBLH, but rather improve the state variable forecast within the PBL.

*· Treatment of observations errors. On L166, it is stated that the observation error variances are equal to the uncertainty estimates provided by the lidar retrievals. In additional to these measurement errors, operational data assimilation systems typically also consider (i) errors of representation and (ii) errors due to approximations in the observation (forward) operator. These additional error contributions might be important to consider in future efforts. For example, errors of representativity might be especially relevant within the more inhomogeneous stable PBL, whereas the methodological differences in the PBLH computation could be treated, at least to a first degree, as errors in the observation operator.*

This is an important issue, and it relates to the different physics schemes since they implicitly represent the forward operator. This is why it is important to conduct assimilation experiments with a number of different schemes, and the resulting impact on the state variable profiles should give a good indication as to whether the ensemble generated forward operator is accurately representing correlations. Some increase in the observation error shoiuld be included here as well. We have added a sentence on this in the Discussion and conclusions section.

*· The ability of PBLH retrievals to efficiently constrain the model state. Here I offer two different comments. The first one concerns the inability of PBLH retrievals to correct the stable PBL structure despite the large differences in the prior and observed PBLH values. The analysis offered by the authors indicates that the lack of ensemble cross-covariances is a likely explanation. This raises the question, however, whether the small corrections in stable PBLs constitutes a systematic effect induced by the formulation of current PBL schemes. Answering this question is important as one of the focal objectives of the PECAN field campaign was to understand if the information provided by ground-based PBL profilers can improve the traditionally poor forecasts of nocturnal convective systems. The second point the authors might want to consider is how effective the PBLH retrievals would be in the context of other observation networks. A past NRC report (NRC 2009) describes the potential deployment of a nation-wide network of thermodynamic and kinematic PBL profilers, quite similar to the ones employed during the PECAN field campaign. Unlike the ceilometer network that originally motivated this study, the PBL profilers will produce direct observations of model state variables that could make the ceilometer-based PBLH retrievals redundant.*

We agree with these concerns. The new plot showing the relative values for $\mathbf{R}$ and $\mathbf{HP}^f\mathbf{H}^T$ over time indicate that the uncertainty in the lidar observations during the night is a big part of the reason. So it would be very helpful to have profile (thermodynamic and kinetic) profiles during this time to complement the PBLH observations. We have added this to the Summary and Conclusions section.

*· My last comment is more technical in nature and relates to the theoretical inappropriateness of current data assimilation systems to extract information from PBLH retrievals. By definition, PBLH is a non-negative quantity. As such, it faces the problems similar to those associated with the assimilation of certain moisture variables (e.g., specific humidity; see Dee and da Silva 2003). Because PBLH is a bounded quantity, its distribution will not always be Gaussian. Hence, traditional assumptions used to derive operational data assimilation schemes will be violated (e.g., Bocquet et al. 2010; Bannister et al. 2020). Such deviations from Gaussianity will be particularly visible under stable PBLs (because PBLH is closer to its lower bound of 0m), further complicating the data assimilation problems already noted in this manuscript. It might be possible to remedy the problem of boundedness by adopting certain non-Gaussian extensions of traditional data assimilation algorithms (cf. Cohn 1997; Fletcher and Zupanski 2006; Bishop 2016).*

PBLH assimilation may benefit from some of the above methods to deal with non-Gaussian statistics. WE have added a sentence to explain this.

*Minor comments*

  *1. L32: Momentum is also exchanged in land-atmosphere interactions, please add to the list.*
done
*2. L46-L47: "... since aerosols are well mixed throughout the PBL (Hicks et al., 2019)" - this statement is only valid in the context of CBL.*

Statement has been changed to address this.

*3. L72-L73: "... were not yet available for the campaign we are using". Before providing details on how the Doppler lidar measurements in Greensburg were obtained, the readers would benefit from a short description of the PECAN field campaign as a whole.*

We have added several sentences to the introduction on the PECAN campaign.

*4. The use of "PBLH retrievals" is more appropriate than "PBLH measurements" as it emphasizes that PBLH is a derived quantity. There are a couple of instances in the manuscript where this correction needs to be applied.*

We have changed this wherever it occurs.

*5. Description of the methods to assimilate the PBLH retrievals (L107-L109). It might be better to replace the statement "either by creating an adjoint of the PBL parameterization scheme" with "either by adopting a variational data assimilation scheme" (or a semantically equivalent expression). Formulating the adjoint of a parameterisation scheme is a specific implementation aspect of variational data assimilation algorithms. If the authors desire so, they could motivate their preference for an EnKF approach in this study by pointing out that ensemble-based methods sidestep the generally difficult task of linearizing the model physics equations.*

We have made this change.

*6. Please cite the original EnKF paper of Evensen (1994) and its subsequent refinement in Burgers et al. (1998) on L123.*

done.

*7. L124: "... where the analysis state is the estimate with the minimum estimated errors". The original derivation of the Kalman filter minimizes the expected squared errors. Note that the latter corresponds to a minimisation of an error norm rather than the error norm implied on L124.*

We have changed the wording here.

*8. The EnOI was originally introduced by Oke et al. (2002) and also discussed by Evensen (2003), so please make sure to add these references on L133.*

Done.

*9. On L141, the authors mention the names of the two parameterization schemes used in the archived NU-WRF simulations. A brief justification on why these two parameterization schemes were chosen in the original PECAN runs will be helpful.*

These simulations were done on a previous project concerned with precipitation that I was involved with. They were chosen because the PBL scheme was the same for both, while the precipitation algorithm is different. It would have been more interesting for this project if we had a different PBL scheme to compare with, but it was not possible. I don't think this information would add any incite to the present work.

*10. The abbreviation TKE on L145 is commonly used to denote "turbulent kinetic energy" in boundary layer research, so it might be best to consider rewording.*

Done

*11. State vector definition (L145-L148). Please clarify whether the state variable $Q$ refers to specific humidity, mixing ratio or another moisture variable. The authors should also stress that the state vector $\mathbf{x}$ in this study represents a vertical column, which is why $P^f$ only refers to the vertical ensemble covariances.*

Done.

*12. Mathematical description of the EnOI algorithm (L160-L191). Here I have a couple of technical remarks aimed at refining the mathematical presentation of the EnOI algorithm.*

*· Because the definition of the forecasted error covariance in Eq. (1) is too general and not specific to the EnOI algorithm, it might be best to remove it from the discussion.*

We agree, since this form is never used in the algorithm. It has been removed.

*· Similarly, it is not necessary to write the measurement equation $y - Hx$; instead, the authors could simply state that $y$ in Eq. (2) represents the PBLH observations retrieved from the Doppler lidar. (As a side comment, the aforementioned measurement equation is only partially complete in the context of filtering theory as it should include a random noise term.)*

We have made this change.

*· It will be best to avoid mixing the vectorial and scalar notations in Eqs. (2) and (3). This could be done by first writing the general form of Eqs. (2)-(3) and then describing how these were solved by the EnOI algorithm employed in this study. Regarding the latter, the authors should highlight that $y$*

as well as the corresponding $HP^fH$ and $R$ matrices are scalar quantities and that both $PfH^T$ and $HP^fH^T$ are computed from an ensemble of model profiles, as indicated by Eqs. (4) and (5).

We have rewritten these equations in the manner you suggested.

· A brief description is needed to explain how the vertical localization mentioned on L177- L185 was implemented in this study.

The vertical covariance is multiplied by the localization factor. We have added this detail.

13. Opening sentences in Section 3 (L193-L199). Some aspects regarding the description of the NU-WRF simulations were already discussed in the paragraph starting on L141. The missing details found on L193-L199 should be moved back to this paragraph.

This has been moved.

14. L205: "... in the late evening to early morning (2-7 UTC)". 7 UTC does not correspond to early morning.

Changed this to "late evening to nighttime"

15. L207: "early morning and early afternoon". Please define this period in the same manner as you have done in other places within the text.

done.

16. Instead of using the temperature as an example in Eq. (7), please use a generic variable, say X, to generalize the RMS difference formula. Moreover, instead of explaining the meaning of i=8 on L220, just mention that the index i denotes a model level.

We think that adding another symbol for "generic variable" is a bit cumbersome at this point in the paper ($\mathbf{x}$ is already representing the entire state vector). The statement about the "top of the PBL" was added at the request of another reviewer. I agree that it is a little too specific. We have changed this as a compromise, but would like to leave in RMS definition using temperature.

17. Statements regarding the corrections made to the WV profile at 22 UTC (L262, L270-L271 and L302-303). The analyzed WV profile overshoots the observed one only with respect to the MYNN simulation. By contrast, the forecasted values in the MYJ WV profiles are already higher, so increasing the WV values following the DA update acts to further increase the observation misfit. Please make sure to make this distinction while describing your results.

We don't quite agree here, though a change is needed. The MYJ WV forecast is high nearer the top of the PBL, but near the surface it is a bit low. The PBLH assimilation only contains a single piece of information, it can only push the profile in one direction, and here it pushed it higher. Increasing the RMS differences. We have changed the text slightly.

18. Justification regarding the deteriorated estimates of the U profile (L266-L268). The authors correctly acknowledge that the estimation of U is a challenging task when one assimilates integrated quantities like PBLH. However, a similar inference can be made in terms of the Vcomponent profile of the wind and, in fact, further argued that the estimation of V is more challenging due to the presence of sharp gradients in the 750-800 hPa layer (cf. lower-right panels in Figs. 6 and 7). An alternative hypothesis to explain the reported differences in the U and V estimates can be linked to the magnitude of the cross-variable ensemble covariances. The lower two panels of Fig. 8, for example, show that the PBLH-V cross-covariances are much larger than their PBLH-U counterparts. Taking into account that the performance of EnOI (and all other ensemble-based DA methods) is especially susceptible to number of ensemble members, it is quite possible that the small PBLH-U covariances are simply a manifestation of the inherent sampling noise, which would in turn act to degrade the quality of the analyzed U profiles. In theory, the above hypothesis could be tested by comparing results with different ensemble sizes (or with different number of vertical model profiles in the context of this study).

This may be a good explanation as well. We don't think that it can be resolved without the implementation of an EnKF. We have added a little along the lines of your suggestions. This is added after the $\mathbf{P}^f\mathbf{H}^T$ plots are introduced.

19. L272-L276: The description of how the PBLH retrievals correct the state variables should be either removed or relocated to the methodology section.

We have removed these lines.

20. L284-287: Here the authors state that the "... more limited velocity corrections are largely constrained by the correlations ...". This is only partially true as Fig. 8 shows that the V crossco-variances are considerably larger than their U counterparts.

This is a little to simplistic an argument. What matters is the relative size of the covariance between nighttime and late afternoon. The magnitude of the temperature and WV covariance increase by

an order of magnitude, whereas the velocity covariances (while changing sign) don't increase in magnitude nearly as much. We have changed this statement slightly in the text.

*21. L311: Replace "covariances" with "ensemble covariances" to emphasize the underlying computational method.*

Done.

*22. L311: It will be best to change "defined" to "controlled". The ensemble covariances are defined mathematically through the sample covariance formula, but controlled by the characteristics of the T, MV, U and V profiles that enter the PBL parameterization schemes.*

Done.

*23. L314: Did you intend to refer to the "analysis profiles" instead of the "forecast profiles" here?*

This should be forecast, and we have changed it.

*Typos and stylistic changes*

*1. The authors should consider segmenting their results in Section 3 with a view of enhancing the readability of their manuscript. One possibility would be to divide the results into three subsections. The first one could discuss the discrepancies between modelled and retrieved PBLH values, the second one – how the assimilated PBLH observations correct the observed and unobserved model variables, while the third might focus on interpreting the magnitude of the T, MV, U and V corrections at 04 UTC and 22 UTC.*

We have divieded this section into 3 subsections as you suggest.

*2. The paragraph starting on L255 could be merged with the preceding one as it provides a summary of the main results.*

Done.

*3. Please correct the spelling of EnKF on L325 and L329.*

Done.

*4. There were several places where "Doppler" was not capitalized.*

This has been corrected.

*5. I spotted unintended word repetitions in a couple of places within the text, e.g. L252, L255 and L303.*

These have been removed.

*6. L23: "leading to an increased differences" – remove "an".*

Done.

*7. L36: "... rapidly transported within this layer". It is not clear which layer is being referred to as the previous sentence mentions both the CBL and SBL.*

changed this to "rapidly transported within the CBL".

*8. L38: "The PBLH is fundamental to ...". This sentence provides a very general description on the significance of PBLH and should be mentioned earlier in the paragraph, e.g. after the sentence starting on L32.*

This sentence was combined with the first sentence in the introduction.

*9. L42: "penetrates the top" could be changed to "penetrates its top" to make it clear that the authors refer to the PBL top.*

Done.

*10. L186: "This system is solved...". Which system? Please be more specific by listing the relevant equations.*

Changed to identify the analysis equations as being solved.

*11. L208: "... the MYJ for (red triangles) both are higher than the observations". Please confirm that "both" refers to the MYJ forecasts in the early morning and early afternoon. If this is the case, remove "both" as it is clear from the context. and 8 UTC) ..."] makes it hard to interpret the sentence starting on L225.*

We have changed this sentence to indicate both "MYJ"(red triangles) and "MYNN" (blue squares) are be referred to.

*13. L232: "increase" should be replaced with "increases".*

Instead we changed "difference" to "differences" and "analysis" to "analyses".

*14. L237: "(0.5m/2 decrease)" should be replaced with "(0.5m/s decrease)".*

Done.

*15. L240: "inthe RMS differences" – please separate "in" from "the".*

done.

*16. L249-L250: "In the late afternoon (Figures 6,7) indicate ..." – please remove the brackets and refine the sentence structure.*

done.

*17. L260: "The WV profile is shown to be increased ...". It is the WV values that increase, not the profile itself.*

Removed "profile".

*18. L270: Remove "in" from "in show that".*

Done.

*19. L279: "We can also analyze this ...". Not clear what "this" refers to, please clarify.*

Changed "this" to "the assimilation".

*20. L321-L322: "will require the construction of an EnkF, and run over many days" – please correct the wording in this sentence.*

We have changed this sentence slightly.

*Figures*

*1. It would be useful to label the figure panels with (a), (b), etc.*

Done.

*2. Please avoid repetitions in the title and axis labels (e.g., potential temperature in the upperleft panel of Fig. 5).*

This has been changed to avoid the repetition.

*3. Legends are sitting atop some of the curves in Figs. 4-7. Please make sure that all data is displayed in the revisited figures.*

We have left in only one legend for each set of 4 plots so that the curves are no longer covered.

*4. Fig. 8: Please include the ensemble covariance units on the x-axis. Please also replace "for PBL physics model MYHH" to "for the MYNN PBL scheme".*

Done.

[revised manuscript text omitted]